# ADAPTIVE TRAINING OF INRS VIA PRUNING AND DENSIFICATION

## ABSTRACT

Encoding input coordinates with sinusoidal functions into multilayer perceptrons (MLPs) has proven effective for implicit neural representations (INRs) of low-dimensional signals, enabling the modeling of high-frequency details. However, selecting appropriate input frequencies and architectures while managing parameter redundancy remains an open challenge, often addressed through heuristics and heavy hyperparameter optimization schemes. In this paper, we introduce AIRe (**A**daptive **I**mplicit neural **Re**presentation), an adaptive training scheme that refines the INR architecture over the course of optimization. Our method uses a neuron pruning mechanism to avoid redundancy and input frequency densification to improve representation capacity, leading to an improved trade-off between network size and reconstruction quality. For pruning, we first identify less-contributory neurons and apply a targeted weight decay to transfer their information to the remaining neurons, followed by structured pruning. Next, the densification stage adds input frequencies to spectrum regions where the signal underfits, expanding the representational basis. Through experiments on images and SDFs, we show that AIRe reduces model size while preserving, or even improving, reconstruction quality. Code and pretrained models will be released for public use.

## 1 INTRODUCTION

Implicit neural representations (INRs) have emerged as a powerful framework for modeling low-dimensional signals – such as images and signed distance functions (SDFs) – by encoding them directly in the parameters of neural networks (Sitzmann et al., 2020; Tancik et al., 2020; Saragadam et al., 2023; Dam et al., 2025). Instead of storing signals discretely, INRs represent them as continuous functions, mapping input coordinates $\mathbf{x}$ to a network predicting the corresponding signal value. To capture high-frequency content, these networks typically employ two key components: (1) projecting $\mathbf{x}$ into a list of sinusoidals $\sin(\omega\mathbf{x} + \varphi)$, where $\omega$ and $\varphi$ denote the input frequencies and phase shifts, and (2) using periodic activation functions throughout the network layers. This combination enables INRs to represent fine details that standard ReLU-based MLPs struggle to learn due to their spectral bias (Tancik et al., 2020; Sitzmann et al., 2020).

Choosing an appropriate network architecture and input frequencies $\omega$ to accurately and compactly fit a target signal is a challenging task. Most prior work has addressed this by enhancing the expressiveness of INRs via tailored initialization schemes and specialized activation functions. For example, Zell et al. (2022) leveraged an initialization based on Fourier series to control the network's spectrum, enhancing its ability to represent fine-grained details. TUNER (Novello et al., 2025) provided a theoretical justification for this approach and introduced a training procedure to bandlimit the spectrum dynamically. FINER (Liu et al., 2024), on the other hand, employed a modified sine activation combined with bias initialization, allowing the modeling of high-frequency components. Despite these advances, selecting a compact yet expressive architecture a priori remains difficult: undersized networks tend to underfit the data, while oversized ones often lead to training instabilities and increased susceptibility to overfitting.

To address this challenge, we introduce **AIRe** (**A**daptive **I**mplicit neural **Re**presentation), a training framework that progressively adapts a potentially overparametrized INR to the target data through two complementary operations: *pruning* and *densification* of neurons. For pruning, we evaluate the contribution of each neuron using a customizable criterion (e.g. weight norms) to identify the

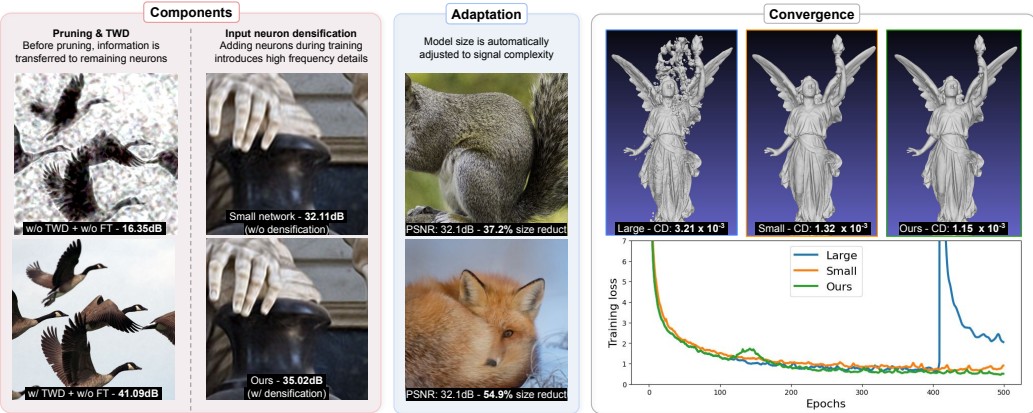

Figure 1: We present AIRe, a robust training method that adaptively fits the INR architecture to the target signal through two complementary mechanisms: (i) pruning with targeted weight decay (TWD) which mitigates parameter redundancy and fine tuning (FT) dependence by transferring information prior to structured neuron removal (see birds), and (ii) input frequency densification, which augments the representation basis, enhancing convergence and details fidelity (see hand). We compare three strategies: (i) an overparameterized SIREN model with standard training (*large network*), (ii) a model adapted with AIRe (**Ours**), and (iii) a *small network* fitted with standard training. AIRe improves reconstruction accuracy while producing more compact networks (blue box), and enhances training convergence in settings where overparameterization leads to divergence (see statue box).

most redundant ones. To transfer information from these low-contributing neurons to more relevant ones, we propose a novel *targeted weight decay* (TWD) mechanism, which penalizes their weights prior to structured removal. Once this transfer is induced, the targeted neurons are pruned. For densification, we introduce new input frequencies in underfit regions of the spectrum, expanding the network's representational capacity when necessary. By dynamically aligning model complexity with the input data, AIRe finds compact INRs that accurately reproduce the target signal. We showcase some of AIRe's results in Figure 1, illustrating strong performance in reconstruction quality, model compactness, and training stability. **Our main contributions are:**

- A general framework for the adaptive training of INRs, driven by pruning and densification. The pruning component brings principles from neural network pruning to the INR setting, while also introducing a novel targeted weight decay (TWD) strategy to preserve quality during neuron removal (see Figure 5). For densification, we add new input frequencies in underfit spectral regions, enhancing representational capacity (Table 4). Combined, these components enable accurate signal fitting with compact, data-adaptive architectures (Table 1).

- A theoretical analysis of both pruning and densification mechanisms for INRs. In particular, we leverage a harmonic expansion of sinusoidal neural networks (Theorem 1) to derive principled densification schemes, and prove stability of our neural networks under magnitude-based pruning (Theorem 2). Together, these promote densification and pruning mechanisms that mitigate divergence during training (cf. Figure 4).

- An empirical evaluation of AIRe across a range of image fitting and 3D shape reconstruction benchmarks. We show that AIRe consistently outperforms both the standard neural network training pipeline (see Table 1) as well as recent adaptive training methods (Table 2) in terms of the accuracy-efficiency trade-off.

## 2 RELATED WORK

**INRs** emerged as a modern paradigm for learning low-dimensional signals such as images (Chen et al., 2021; Shi et al., 2024; Shah and Sitawarin, 2023; Kania et al., 2024; Han et al., 2025; Lee et al., 2021; Jayasundara et al., 2025), image morphing (Schardong et al., 2024; Bizzi et al., 2025), SDFs (Yang et al., 2021; Novello et al., 2022; Schirmer et al., 2024), displacement fields (Yifan et al.,

2021), surface animation (Mehta et al., 2022; Novello et al., 2023), and multiresolution signals (Paz et al., 2023; Saragadam et al., 2022; Lindell et al., 2022; Wu et al., 2023). On the methodological side, several works have explored the representation capacity of INRs (Mehta et al., 2021; Yüce et al., 2022; Saratchandran et al., 2025), as well as the critical role of initialization strategies (Novello, 2022; Paz et al., 2024; Saratchandran et al., 2024; Finn et al., 2017; Yeom et al., 2024).

**Neural network pruning** has long been of interest to the machine learning community (LeCun et al., 1989; Hassibi et al., 1993; Thimm and Fiesler, 1995; Frankle and Carbin, 2019; Hoefler et al., 2021; Blalock et al., 2020; Menghani, 2023). Classic approaches have relied on metrics such as weight magnitude, salience, or second-order derivatives, and are often followed by fine-tuning or regularization (e.g., weight decay) to preserve performance (Han et al., 2015; Tessier et al., 2022). However, it is known that methods often fail to generalize beyond their original settings (Blalock et al., 2020). To the best of our knowledge, Zell et al. (2022) is the only prior work exploring the pruning (or adaptation) of INRs. Their method removes input neurons to select an appropriate representational basis, but they did not explore hidden neurons pruning. In contrast, our method adapts the model size to target redundancy in the signal detail content while choosing a fitting input frequency encoding.

Recent work has investigated ways to adapt network architectures during training. The lottery ticket hypothesis (Frankle and Carbin, 2019) suggests that sparse subnetworks within overparameterized models can perform just as well when trained independently. Building on this idea, RigL (Evci et al., 2020) dynamically adjusts connectivity by pruning and growing connections during training. Also of note is Gaussian Splatting and related works (Kerbl et al., 2023; Zhang et al., 2024; Waczyńska et al., 2024), whose training procedures are adaptive and feature densification and pruning mechanisms. Nevertheless, while promising, such strategies have not been studied in the context of INRs, where the objectives, data modalities, and inductive biases differ significantly from those in previous endeavors. In Table 2, we adapt these methods to the INR setting and compare them with AIRe, showing that our approach achieves superior results.

## 3 ADAPTIVE TRAINING OF INRS

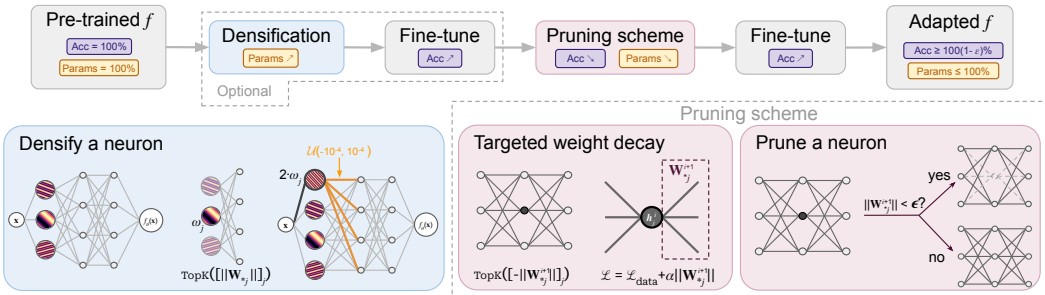

Figure 2: We present AIRe, a training framework that adapts network architecture through two theoretically grounded strategies: *densification* and *pruning*. For signals with rich frequency content, densification selects the most relevant input frequencies $\omega_j$ and expands the spectrum by augmenting $\omega$ with $2 \cdot \omega_j$. To reduce network size, pruning identifies candidate neurons via magnitude criterion, transfers information during training with a novel targeted weight decay (TWD) regularization, and removes neurons whose norm falls below a threshold $\epsilon$. The function TopK($v$) selects the $K$ largest entries of $v$.

Our goal is to develop a training framework that dynamically adapts a sinusoidal INR architecture to the given data samples $\{\mathbf{x}_j, f_j\}$ from a low-dimensional signal $f$. Specifically, we want to adjust the size of a sinusoidal MLP of depth $d \in \mathbb{N}$ defined as $f(\mathbf{x}) = \mathbf{L} \circ \mathbf{S}^d \circ \cdots \circ \mathbf{S}^0(\mathbf{x})$, a composition of $d$ sinusoidal layers $\mathbf{S}^i(\mathbf{x}) = \sin(\mathbf{W}^i \mathbf{x} + \mathbf{b}^i)$ parameterized by a weight matrix $\mathbf{W}^i \in \mathbb{R}^{n_{i+1} \times n_i}$ and a bias vector $\mathbf{b}^i \in \mathbb{R}^{n_{i+1}}$, followed by an affine layer $\mathbf{L}$. Observe that the first layer $\mathbf{S}^0$ maps the input coordinates $\mathbf{x}$ into a harmonic embedding of the form $\sin(\omega \mathbf{x} + \varphi)$, where we denote $\omega := \mathbf{W}^0$ as the matrix of *input frequencies* and $\varphi := \mathbf{b}^0$ as the vector of *phase shifts*.

Although the choice of $\{n_i\}_i$ is critical for determining network capacity, it is typically based on empirical heuristics. Moreover, a model with poorly initialized input frequencies $\omega$ may fail to

capture the full spectrum of the signal, leading to unsatisfactory reconstruction. To address these problems, we adapt a model architecture by adding and removing neurons. More precisely, we define the $ij$-neuron $h_j^i(\mathbf{x})$ of $f$ as the $j$-th coordinate of the output of the $i$-th layer, that is,

$$h_j^{i+1}(\mathbf{x}) = \sin\big(\mathbf{W}_{j*}^{i+1} \sin(\mathbf{y}^i) + b_j^{i+1}\big), \tag{1}$$

where $\mathbf{y}^i$ denotes the linear transformation of the $i$th layer prior to activation. Then, we densify the input neurons by appending new ones to $\mathbf{h}^0(\mathbf{x})$, introducing novel frequencies to expand the spectral coverage. Finally, we employ a magnitude-based neuron pruning scheme to account for potential redundancy in parameters. Figure 2 provides an overview of AIRe.

## 3.1 DENSIFICATION

Sinusoidal INRs employ an embedding layer parametrized by input frequencies $\omega$ which are passed through the network, generating additional frequencies, enriching the network's spectrum so as to mitigate spectral bias. However, the representation capacity of such networks is heavily dependent on the initialization of $\omega$, which may lead to noisy reconstructions or slower training when done poorly. To mitigate this, we introduce a densification strategy for the set of input frequencies $\omega$ that identifies important generated frequencies (those with high amplitude in a certain trigonometric expansion of the network) and introduce them directly in the input frequencies $\omega$, enriching the network.

To design this densification strategy, we must analyze the network spectrum. This can be done by a theorem of Novello et al. (2025), which provides a trigonometric expansion that facilitates this analysis.

**Theorem 1.** *The neuron $h_j^{i+1}$ admits the following amplitude-phase expansion:*

$$h_j^{i+1}(\mathbf{x}) = \sum_{\mathbf{k}\in\mathbb{Z}^{n_i}} \alpha_{\mathbf{k}}(\mathbf{W}_{j*}^i) \, \sin\big(\langle \mathbf{k}, \mathbf{y}^i\rangle + b_j^{i+1}\big), \quad \text{where} \quad |\alpha_{\mathbf{k}}(\mathbf{W}_{j*}^i)| \le \prod_l \frac{1}{|k_l|!}\left(\frac{|W_{jl}^{i+1}|}{2}\right)^{|k_l|} \tag{2}$$

*Here, $\alpha_{\mathbf{k}}(\mathbf{W}_{j*}^i) = \prod_l J_{k_l}(W_{jl}^{i+1})$ is the product of Bessel functions.*

This result shows that the composition of sinusoidal layers generates new frequencies of the form $\langle \mathbf{k}, \omega\rangle$, depending solely on the input frequencies $\omega$, with phase shifts determined by the biases $\{\varphi, \mathbf{b}^i\}$. Additionally, the amplitudes $\alpha_{\mathbf{k}}$ depend exclusively on the hidden weight matrices $\mathbf{W}^i$. Thus, the generated frequencies are governed by the input embedding, while the hidden parameters control the amplitudes and phase shifts. Moreover, from Equation 2 we observe that

$$\mathbf{h}^0(\mathbf{x}) = \left[\sum_{\mathbf{k}\in\mathbb{Z}^{n_0}} \alpha_{\mathbf{k}} \sin\left(\langle \mathbf{k}, \omega\rangle\mathbf{x} + b_j\right)\right]_j \quad \text{with bias } b_j = \langle \mathbf{k}, \varphi\rangle + b_j^1. \tag{3}$$

Thus, adding an input neuron with frequency $\omega'$ expands the layer spectrum from $\{\langle \mathbf{k}, \omega\rangle\}_{\mathbf{k}}$ to $\{\langle \mathbf{k}, \omega\rangle + l \cdot \omega'\}_{\mathbf{k},l}$. Since the input frequencies determine the network's generated frequencies, the densification of the input neurons can greatly increase the expressiveness of the overall network.

Equation 3 suggests a natural set of new input frequencies $\omega'$ to be added: multiples of input frequencies whose corresponding columns in the first hidden layer have high norm. Specifically, if the $j$th column $W_{*j}^1$ of the first hidden layer matrix $\mathbf{W}^1$ has a high norm, then the lowest multiples of its associated frequency $\omega_j$ tend to appear with larger amplitudes.[1] Therefore, during densification, we identify input frequencies $\omega_j$ whose corresponding columns $W_{*j}^1$ exhibit high norms, and we expand the set of input frequencies $\omega$ by adding their doubled counterparts $2\omega_j$, boosting the input frequencies and thus enriching the network's representation. As shown in Figure 10, after densification most of the newly added neurons become highly contributive to the reconstruction.

Finally, the corresponding new column in the hidden matrix $\mathbf{W}^1$ is initialized with random values drawn from a uniform distribution in the range $[-10^{-4}, 10^{-4}]$, ensuring a stable start for training. Finally, the network is retrained to fine-tune all parameters, allowing it to adapt to the extended frequency spectrum and fully leverage the increased representational capacity.

---

[1] This follows from Equation 3 and the fact that $|J_2(W_{ij}^1)| > |J_{k_j}(W_{ij}^1)|$ for most $i$, since Bessel functions are sorted by their order (Novello, 2022, Sec 2.2).

## 3.2 PRUNING

Determining an appropriately sized model capable of representing the target signal with quality is a key challenge when training sinusoidal INRs. Typically, large architectures are employed to ensure reconstruction accuracy, sacrificing model compactness. To address this, we design a pruning procedure that detects and removes redundant neurons during training.

First, we employ a magnitude-based criterion to identify uninformative neurons, a common strategy in classical network pruning. We now provide a formal justification of its validity for INRs: in sinusoidal MLPs, pruning neurons induces only a bounded perturbation to the overall function. This perturbation depends on the $\infty$-operator norms of the parameter changes and the norms of the subsequent layers.

**Theorem 2.** *Let $f$ be a sinusoidal INR of depth $d$, and let $\widetilde{f}$ be the network obtained by perturbing the $k$-th hidden layer weights and biases to $\widetilde{\mathbf{W}}^k$ and $\widetilde{\mathbf{b}}^k$. Then,*

$$\sup_x \left\| f(x) - \widetilde{f}(x) \right\|_\infty \leq \left( \left\| \mathbf{W}^k - \widetilde{\mathbf{W}}^k \right\|_\infty + \left\| \mathbf{b}^k - \widetilde{\mathbf{b}}^k \right\|_\infty \right) \|\mathbf{L}\|_\infty \prod_{i=k+1}^{d} \|\mathbf{W}^i\|_\infty.$$

Theorem 2 formally guarantees that small modifications to a layer's parameters—such as pruning neurons with small outgoing weights—induce only proportionally small changes to the network's output. This justifies magnitude-based pruning both intuitively and theoretically. However, training directly with the reconstruction loss $\mathcal{L}_{\text{data}}$ (e.g. MSE) often leads to relatively few truly redundant neurons, even in overparametrized architectures. For better pruning, we employ a targeted weight decay (TWD) strategy that reduces the contribution from near-redundant neurons, turning them truly redundant. It consists of training the network $f$ with the loss function,

$$\mathcal{L}_{\alpha,\mathcal{I}} = \mathcal{L}_{\text{data}} + \alpha \sum_{j \in \mathcal{I}} \|\mathbf{W}^{i+1}_{*j}\|_1, \quad \text{with} \quad \alpha \in [0, 1), \tag{4}$$

where $\mathcal{I} = \text{TopK}\left( \left[ -\|\mathbf{W}^{i+1}_{*j}\|_1 \right]_j \right)$ are the $K$ indices of the neurons with the smallest column norm.

Our procedure uses TWD to isolate low-impact neurons, ensuring that pruning remains consistent with the theoretical stability. Then, we select the neurons to remove by thresholding small $\ell_1$ norms, e.g., pruning $h^i_j$ if $\|\mathbf{W}^{i+1}_{*j}\|_1 = \|\mathbf{W}^{i+1} - \widetilde{\mathbf{W}}^{i+1}\|_\infty \leq \epsilon$ (where $\widetilde{\mathbf{W}}^{i+1}$ denotes the altered weight matrix), and fine-tune the network to recover performance. As illustrated in Figure 2, pruning a neuron $h^i_j(\mathbf{x})$ involves removing its outgoing connections. In practice, we mask only the entries of the $j$-th column $\mathbf{W}^{i+1}_{*j}$, which implicitly leaves unused the row $\mathbf{W}^i_{j*}$ and bias $b^i_j$. Note that pruning the input neurons may have greater impact on the reconstruction since we are deleting an input frequency; that is, we are eliminating many generated frequencies from the network spectrum.

## 4 EXPERIMENTS

We evaluate AIRe on adaptive training across three tasks: image fitting, surface reconstruction (SDFs), and novel view synthesis with NeRFs. Experiments are conducted on the DIV2K (Agustsson and Timofte, 2017), Stanford Repository (Curless and Levoy, 1996), and NeRF Synthetic (Mildenhall et al., 2021) datasets. We also study AIRe in a setup where the final architecture is fixed, demonstrating that our training procedure can improve reconstruction quality even when the reduced small architecture is known in advance. Finally, we perform ablation studies to validate the design choices underlying our method.

All models are implemented in PyTorch (Paszke et al., 2019) and optimized with Adam (Kingma and Ba, 2015).For simplicity, we denote a MLP by $[n_1, ..., n_{d+1}]$, where $d$ is the number of hidden layers and $n_i$ is the number of neurons in the $i$-th layer. The pruning threshold is set to $\epsilon = 0.01$, which prevents neurons with contributions above this value from being pruned; this threshold is fixed and applied uniformly across all layers.

**Comparison with standard training.** We compare AIRe against a baseline defined by the original, large initial architecture (overparametrized) trained with the standard neural network training pipeline,

showing that AIRe can reduce model size while maintaining reconstruction quality by finding more appropriate input frequencies. We evaluate this on images, SDFs, and NeRFs, adopting commonly used architectures for each task (SIREN and FINER). Table 1 shows that AIRe achieves substantial reductions in SIREN's model size while maintaining reconstruction quality, and in several cases even improving it.

Table 1: **Our method fits a compact INR to the target signal while preserving accuracy.** We evaluate AIRe ('Ours') against an overparametrized INR trained with the standard training pipeline ('Large') on images (with model size $[512, 256, 256]$), SDFs (with architecture $[256, 256, 256]$), and NeRF tasks (with size $[256, 128, 128]$), reporting PSNR and Chamfer Distance ($\times 10^2$). AIRe enables a strong reduction in model size, while preserving or even improving quality.

| Imgs (DIV2K) | Large PSNR↑ | Ours PSNR↑ | Size reduct.↑ | SDFs (Stanford) | Large CD↓ | Ours CD↓ | Size reduct.↑ | NeRF (Synthetic) | Large PSNR↑ | Ours PSNR↑ | Size reduct.↑ |
|---|---|---|---|---|---|---|---|---|---|---|---|
| #00 | 31.96 | 31.56 | 35.89% | Armadillo | 0.62 | 0.63 | 73.30% | Lego | 25.72 | 25.30 | 35.00% |
| #01 | 37.93 | 35.63 | 65.28% | Bunny | 0.76 | 0.71 | 72.33% | Materials | 23.71 | 23.62 | 30.86% |
| #02 | 30.76 | 29.17 | 52.39% | Dragon | 0.73 | 0.61 | 70.33% | Ficus | 24.23 | 24.82 | 26.83% |
| #03 | 37.40 | 35.04 | 56.80% | Buddha | 0.59 | 0.56 | 41.27% | Hotdog | 29.69 | 28.68 | 33.14% |
| #04 | 33.88 | 31.09 | 60.08% | Lucy | 0.92 | 0.58 | 52.50% | Drums | 22.17 | 22.02 | 39.09% |

**Comparison against existing pruning baselines.** Table 2 reports a comparison with existing pruning approaches on the image representation task, using the same SIREN configuration as in Table 1. We evaluate two model-agnostic pruning methods, DepGraph (Fang et al., 2023) and RigL (Evci et al., 2020), as well as a baseline obtained by training a small architecture from scratch. We also compare against Zell et al. (2022), a method that initializes the first SIREN layer with a wide range of frequencies given by the Cartesian product between

Table 2: **Comparison of pruning criteria.** Results are on the image representation task.

| Method | PSNR↑ | SSIM↑ |
|---|---|---|
| Small | 34.60 ± 3.82 | 0.92 ± 0.03 |
| DepGraph | 27.56 ± 2.12 | 0.82 ± 0.04 |
| RigL | 34.29 ± 3.37 | **0.95 ± 0.01** |
| Zell et al. (2022) | 18.96 ± 1.92 | 0.39 ± 0.08 |
| AIRe (ours) | **37.07 ± 3.74** | **0.95 ± 0.01** |

$[0, 1, \dots, L]$ and $[-L, \dots, L]$ for a chosen maximum frequency $L$. In our experiment, we set $L = 19$, yielding 780 input frequencies—substantially larger than the first-layer width of 512 used in AIRe. We train their model for 5,000 epochs and prune the input frequencies following the strategy described in their paper. The pruning rate of each method is set to approximately 25% of the original parameters, and we follow the hyperparameter choices reported in the respective papers. AIRe consistently outperforms these pruning methods, demonstrating its effectiveness for INR architecture adaptation over training.

## 4.1 AIRe vs. small networks

AIRe starts with a **large** architecture and progressively reduces its size during training, resulting in a **small** network. To evaluate how effectively AIRe leverages its architectures, we compare it against standard training applied directly to both the initial (large) architecture and the final (small) one. We conduct this evaluation for image fitting (DIV2K) and SDF reconstruction (Stanford Repository).

For the SDF reconstruction task, we follow the implementation in (Novello et al., 2022), training each network for $10^3$ epochs, sampling $10^4$ on-surface points and $10^4$ off-surface points uniformly. Meshes are extracted from the trained SDFs via marching cubes with a resolution of $512^3$, and all surfaces were normalized to $[-1, 1]^3$. For evaluation, we report the number of network parameters (Params) and the Chamfer Distance (CD) between reconstructed and ground-truth surfaces. We also evaluate AIRe without densification, as SDFs typically contain less details than other applications.

For training, we start with a large network $[256, 256, 256]$, trained from scratch for 200 epochs. We then select 192 neurons from the input layer and the first hidden layer and continue training with targeted weight decay (TWD) for 500 epochs. Finally, the selected neurons are pruned, and the resulting smaller network $[64, 64, 256]$ is retrained for 300 epochs. Table 3(left) shows that AIRe provides a better SDF reconstruction than the large network in all cases. We also highlight that our

Table 3: **AIRe vs. directly trained large and small networks.** We compare AIRe with standard training applied to large and small architectures on both SDF reconstruction (Stanford) and image fitting (DIV2K). Metrics are CD ($\times 10^2$) for SDFs and PSNR for images, along with parameter reduction relative to the large model. AIRe achieves accuracy comparable to or better than the large network while using the same reduced parameter budget as the small one.

| Model (SDFs) | Variant | CD ($\times 10^2$) ↓ | Size reduct. ↑ | Model (Images) | Variant | PSNR ↑ | Size reduct. ↑ |
|---|---|---|---|---|---|---|---|
| | Large | $0.65 \pm 0.11$ | - | | Large | $\mathbf{39.59 \pm 3.30}$ | - |
| SIREN | Small | $0.89 \pm 0.09$ | 83.96% | SIREN | Small | $34.60 \pm 3.82$ | 24.95% |
| | Ours | $\mathbf{0.64 \pm 0.03}$ | 83.96% | | Ours | $37.07 \pm 3.74$ | 24.95% |
| | Large | $2.14 \pm 0.41$ | - | | Large | $38.77 \pm 2.98$ | - |
| FINER | Small | $5.08 \pm 3.51$ | 83.96% | FINER | Small | $38.87 \pm 3.44$ | 24.95% |
| | Ours | $\mathbf{0.88 \pm 0.15}$ | 83.96% | | Ours | $\mathbf{39.91 \pm 3.89}$ | 24.95% |

approach obtains similar or better accuracy compared to the initial, large network trained from scratch while using roughly between a third and a sixth of the network parameters for surface representation. Aditionally, we found that AIRe has a comparable time overhead ($76.8s$) compared to the large SIREN model ($76.0s$). Moreover, even when training a small architecture during $83.2s$, it performs worse than AIRe with $0.86 \times 10^2$ for CD metric. Figure 3 shows some qualitative comparisons of AIRe and the small architecture with standard training on the Armadillo, Buddha, and Lucy models, showing that AIRe provides, in general, a lower (bluer) distance from the ground-truth surface.

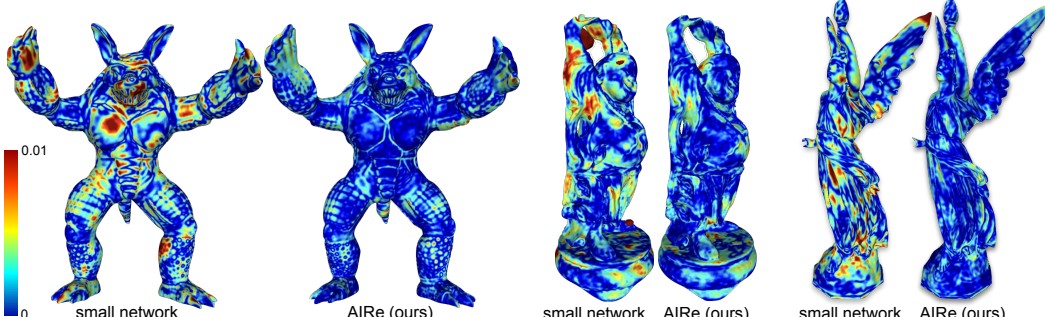

Figure 3: **Qualitative comparison of SDF reconstructions** on the Armadillo, Buddha, and Lucy models using a SIREN with $\omega_0 = 60$ and small network size $[64, 64, 256]$. Left: results of training the final small network directly. Right: results of AIRe. Colors indicate the distance from the ground-truth surface, from dark blue (0) to dark red ($\geq 0.01$). AIRe produces reconstructions that are consistently closer to the ground truth than those obtained by training the small network from scratch.

Additionally, AIRe mitigates divergence during the training of SDF models, as illustrated in Figure 4. We illustrate this by initializing a large network of size $[256, 256, 256, 256]$ and training it on the Armadillo for half the epochs with $\omega_0 = 60$ and small network architecture $[64, 64, 64, 256]$. Under standard training, the large network diverges, producing reconstructions with severe noise and artifacts. In contrast, AIRe yields a more accurate reconstruction despite using less than half the parameters of the initial model.

For the evaluation on the image fitting task, we use the FINER subset of the DIV2K dataset, randomly selecting 90% of the pixels of each image for training and using the remaining 10% for testing. Training is performed with Mean Square Error (MSE) loss, a batch size of 65,536 pixels, and evaluation is based on Peak Signal-to-Noise Ratio (PSNR). All experiments use sinusoidal MLPs with $\omega_0 = 30$ trained for 5000 epochs. For AIRe, we first train for 250 epochs with MSE to capture low-frequency information, then add 128 new input neurons and fine-tune for 2000 epochs. Next, we train for 2250 epochs with TWD, prune 384 input neurons, and fine-tune the resulting network for an additional 500 epochs. The resulting small network of size $[256, 512, 512]$ is compared against a model of the same size trained from scratch with MSE for 5000 epochs.

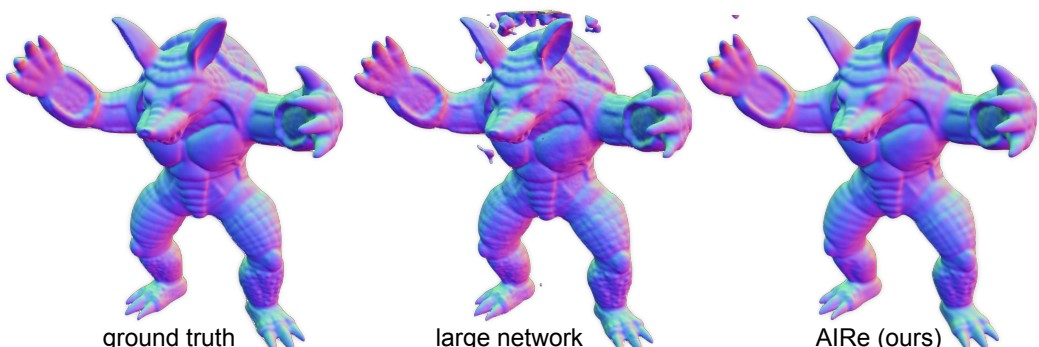

Figure 4: Qualitative comparison on the Armadillo. Left: ground truth. Middle: standard training of the large network, which diverges and produces noisy artifacts. Right: AIRe, which avoids divergence and yields a cleaner reconstruction with fewer parameters.

Table 3 (right) shows that AIRe applied to SIREN and FINER networks achieves better convergence than standard training applied directly to either large or small networks. AIRe improves mean accuracy by 2.47 dB on SIREN and 1.04 dB on FINER, consistently outperforming standard MSE training. These results demonstrate that AIRe effectively transfers information from the overparameterized model to its small counterpart.

## 4.2 ABLATIONS

**Effect of varying pruning rate.** We now ablate key design choices of AIRe, focusing on the rate of pruning and densification during training. First, we analyze the effect of pruning on reconstruction quality. We compare the accuracy drop of an adapted INR relative to a pre-trained network of size $[512, 512, 512]$ (528K parameters), which achieves a PSNR of 43.67 dB. We apply AIRe to the same architecture, starting with standard training for 2250 epochs, followed by selecting $p\%$ of the input neurons and the first hidden neurons to prune ($p \in \{0.2, 0.4, 0.6, 0.8\}$) and training with TWD for another 2250 epochs. Finally, we prune the selected neurons and fine-tune the resulting network for 500 epochs, totaling 5000 epochs of adaptation.

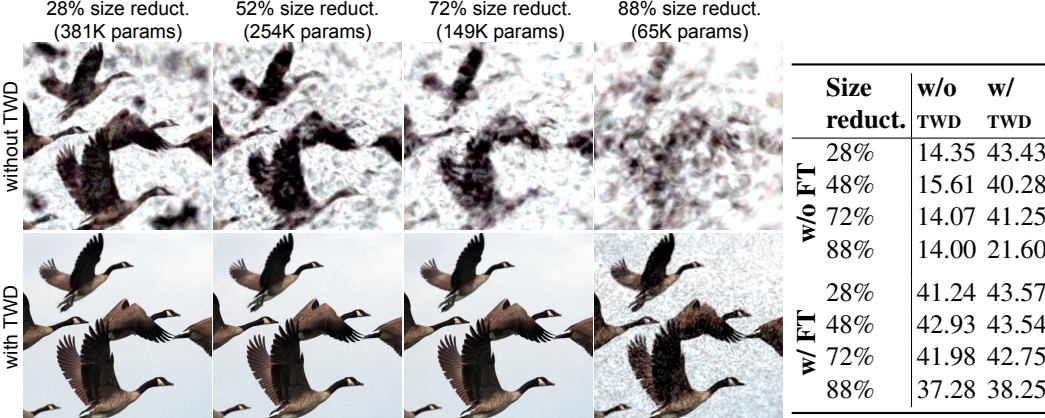

| | Size reduct. | w/o TWD | w/ TWD |
|---|---|---|---|
| **w/o FT** | 28% | 14.35 | 43.43 |
| | 48% | 15.61 | 40.28 |
| | 72% | 14.07 | 41.25 |
| | 88% | 14.00 | 21.60 |
| **w/ FT** | 28% | 41.24 | 43.57 |
| | 48% | 42.93 | 43.54 |
| | 72% | 41.98 | 42.75 |
| | 88% | 37.28 | 38.25 |

Figure 5: **TWD reduces the dependence of finetune (FT) when pruning.** TWD effectively transfers information before pruning. Left: qualitative results with $28\%, 52\%, 72\%,$ and $88\%$ of parameters pruned. The first row shows results without TWD, and the second row with TWD. Right: Table with the PSNR values for each case.

As shown in Figure 5, TWD enables effective transfer of information to the remaining neurons so that a pruned network (without fine-tuning) with only $28\%$ of its weights still retains $92\%$ of the original network's accuracy. In contrast, pruning without TWD leads to a severe degradation in image quality.

After full training, AIRe achieves a quality drop of less than $2.1\%$ with just $28\%$ of the parameters, compared to a $3.8\%$ drop when TWD is removed from the pipeline.

**With vs. without densification.** We ablate the role of densification in our pipeline using the same configuration as Table 2, training all models for $5000$ epochs on a subset of DIV2K. In Table 4, we compare: (i) a small architecture trained from scratch; (ii) a large model pruned (without densification) to match the small architecture; and (iii) our proposed AIRe scheme, which iteratively adds and removes

Table 4: **Effect of pruning and densification** on SIREN and FINER networks (DIV2K).

| Method | SIREN PSNR ↑ | FINER PSNR ↑ |
|---|---|---|
| Small | 36.44 ± 4.20 | 40.84 ± 3.85 |
| Prune | 37.58 ± 3.77 | 41.80 ± 3.80 |
| Densify+Prune | **39.47 ± 4.31** | **41.88 ± 4.24** |

input neurons until matching the small architecture. Pruning alone yields a modest accuracy gain for SIREN, while the *Densify+Prune* (AIRe) strategy provides a substantial boost. For FINER, pruning slightly improves reconstruction quality, but densification brings little benefit – consistent with the fact that FINER models are already more expressive and less dependent on additional frequency capacity.

**Pruning after densification vs. before densification.** Intuitively, increasing model capacity before removing redundancies should improve convergence. To verify this, we train a network of size $[256, 512, 256]$ and compare two schedules: pruning before densification and pruning after densification. For *densify before prune*, we train for $400$ epochs with MSE, add $128$ input neurons, fine-tune for $200$ epochs, train with TWD for $200$ epochs, prune $50\%$ of the first hidden neurons, and fine-tune for $1200$ epochs. For *pruning before densify*, we train for $200$ epochs with MSE, continue for $200$ epochs with TWD, prune $50\%$ of the first hidden neurons, fine-tune for $200$ epochs, add $128$ neurons, and finally fine-tune for $1400$ epochs.

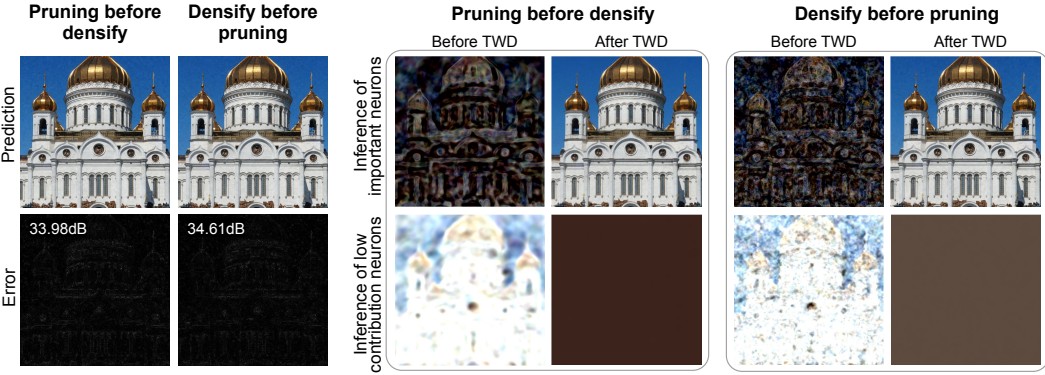

Figure 6: **Effect of pruning-before-densify and densify-before-pruning strategies.** Left: reconstructed signal and error map for each strategy. Right: information transfer during TWD. We visualize the model's inference when restricted to high-contribution neurons (top) and to low-contribution neurons (bottom), evaluated both before and after TWD. TWD offers strong information transfer in both configurations; however, in the densify-before-pruning case, the important neurons capture higher-frequency components, resulting in improved reconstruction quality.

The *pruning-before-densify* strategy achieves an average PSNR of 33.41 dB, whereas *densify-before-pruning* reaches 33.99 dB, confirming our intuition about the ordering of our components. Figure 6 (left) shows a prediction alongside its corresponding error map. On the right, we compare two inferences: the top row uses only parameters of neurons identified as important, while the bottom row uses only the weights of neuron deemed low-contribution. The important-neuron inference preserves most image details, preserving most of the spectral content. In contrast, the inference restricted to low-contribution neurons lacks detail and collapses toward a constant function durint TWD. Note how important neurons in the *densify-before-pruning* scheme contain more high frequency details than its counterpart in the *pruning-before-densify* strategy before the TWD.

## 5 CONCLUSION

We introduced AIRe, a dynamic training framework for implicit neural representations (INRs) that adaptively aligns network architecture with the complexity of the target signal. The framework integrates two complementary components: *pruning*, which removes redundant neurons to mitigate overparameterization, and *densification*, which expands the network's expressivity by selectively introducing new input frequencies based on a principled spectral analysis.

Our approach contributes toward automating architecture adaptation in INR learning, offering a more efficient and flexible alternative to static design choices. As future work, we aim to develop more advanced mechanisms for information transfer during pruning, extend our method to a broader class of architectures, and explore its applicability to more data modalities beyond images and surfaces.

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

# A  Proofs

## A.1  Theorem 1

In the main paper, we present an identity (Thrm 1) derived by Novello et al. (2025), which linearizes the $j$-th hidden neuron of the $(i + 1)$-th layer, $h_j^{i+1}$. Similar results have been presented in Yüce et al. (2022) for the case of shallow SIRENs. The identity below extends this analysis to hidden neurons at arbitrary depths.

**Theorem 3.** *The hidden neuron $h_j^{i+1}$ admits the following amplitude-phase expansion:*

$$
h_j^{i+1}(\mathbf{x}) = \sum_{\mathbf{k} \in \mathbb{Z}^{n_i}} \alpha_{\mathbf{k}} \sin\left(\langle \mathbf{k}, \mathbf{y}^i \rangle + b_j^{i+1}\right), \qquad \text{where} \quad |\alpha_{\mathbf{k}}| \le \prod_l \left(\frac{|W_{jl}^{i+1}|}{2}\right)^{|k_l|} \frac{1}{|k_l|!}. \quad (5)
$$

*Here, $\alpha_{\boldsymbol{k}} = \prod_{l=1}^{n_i} J_{k_l}(W_{jl}^{i+1})$ is the product of Bessel functions.*

Before starting the proof, recall that we defined $h_j^{i+1}(\mathbf{x}) = \sin\left(\sum_{l=1}^{n_i} W_{jl}^{i+1} \sin(y_l^i) + b_j^{i+1}\right)$, with $\mathbf{y}^i = \left[y_l^i\right]_l$ the linear, non-activated contribution of the $i$-th layer. To simplify notation, we drop the indices $i$ and $i + 1$ from $\mathbf{W}^{i+1}, \mathbf{b}^{i+1}, \mathbf{y}^i$, and $n_i$.

*Proof.* The first part of the proof consists of verifying

$$
\begin{aligned}
\sin\left(\sum_{l=1}^{n} W_{jl} \sin(y_l) + b_j\right) &= \sum_{\mathbf{k} \in \mathbb{Z}^m} \alpha_{\mathbf{k}} \sin\left(\langle \mathbf{k}, \mathbf{y} \rangle + b_j\right) \quad \text{and} \\
\cos\left(\sum_{l=1}^{n} W_{jl} \sin(y_l) + b_j\right) &= \sum_{\mathbf{k} \in \mathbb{Z}^m} \alpha_{\mathbf{k}} \cos\left(\langle \mathbf{k}, \mathbf{y} \rangle + b_j\right).
\end{aligned}
\quad (6)
$$

The proof is by induction in $n$. For the **base** case $n = 1$, we use the sum of angles identities and the Bessel function of the first kind properties (see (Abramowitz and Stegun, 1964, num. 9.1.42, 9.1.43)) to prove $\sin\left(W_{j1}\sin(y_1) + b_j\right) = \sum_{k\in\mathbb{Z}} J_k(W_{j1})\sin(ky_1 + b_j)$:

$$
\begin{aligned}
\sin\left(W_{j1}\sin(y_1) + b_j\right) &= \sin\left(W_{j1}\sin(y_1)\right)\cos(b_j) + \cos\left(W_{j1}\sin(y_1)\right)\sin(b_j) \\
&= \sum_{k\in\mathbb{Z}\text{ odd}} J_k(W_{1j})\sin(ky_1)\cos(b_j) + \sum_{l\in\mathbb{Z}\text{ even}} J_l(W_{1j})\cos(ly_1)\sin(b_j) \\
&= \sum_{k\in\mathbb{Z}\text{ odd}} J_k(W_{j1})\sin(ky_1 + b_j) + \sum_{l\in\mathbb{Z}\text{ even}} J_l(W_{j1})\sin(ly_1 + b_j) \\
&= \sum_{k\in\mathbb{Z}} J_k(W_{j1})\sin(ky_1 + b_j).
\end{aligned}
$$

In the third equality we combined the formula $\sin(u)\cos(v) = \frac{\sin(u+v)+\sin(u-v)}{2}$ and the fact that $J_{-k}(u) = (-1)^k J_k(u)$ to rewrite the summations. The proof of the cosine analogous expansion $\cos\left(W_{j1}\sin(y_1) + b_j\right) = \sum J_l(W_{j1} + b_j)\cos(ly_1)$ is similar.

Assume that equation 6 hold for $n - 1$, with $n > 1$, we prove that it also holds for $n$ (the **induction step**).

$$
\begin{aligned}
\sin\left(\sum_{l=1}^n W_{jl}\sin(y_l) + b_j\right) &= \sin\left(\sum_{l=1}^{n-1} W_{jl}\sin(y_l) + b_j\right)\cos\left(W_{jn}\sin(y_n)\right) \\
&\quad + \cos\left(\sum_{l=1}^{n-1} W_{jl}\sin(y_l) + b_j\right)\sin\left(W_{jn}\sin(y_n)\right) \\
&= \sum_{\mathbf{k}\in\mathbb{Z}^{n-1},\, l\in\mathbb{Z}\text{ even}} \alpha_{\mathbf{k}} J_l(W_{jn})\sin\left(\langle \mathbf{k}, \mathbf{y}\rangle + b_j\right)\cos(ly_n) \\
&\quad + \sum_{\mathbf{k}\in\mathbb{Z}^{n-1},\, l\in\mathbb{Z}\text{ odd}} \alpha_{\mathbf{k}} J_l(W_{jn})\cos\left(\langle \mathbf{k}, \mathbf{y}\rangle + b_j\right)\sin(ly_n) \\
&= \sum_{\mathbf{k}\in\mathbb{Z}^n} \alpha_{\mathbf{k}} \sin\left(\langle \mathbf{k}, \mathbf{y}\rangle + b_j\right)
\end{aligned}
$$

We use the induction hypothesis in the second equality and an argument similar to the one used in the base case to rewrite the harmonic sum. Again, the cosine activation function case is analogous.

For the second part of the proof, we must prove the inequality in Equation equation 1. For that, note that $\alpha_{\mathbf{k}} = \prod_{l=1}^n J_{k_l}(W_{jl})$, and that

$$
|J_k(W_{jl})| < \frac{\left(\frac{|W_{jl}|}{2}\right)^k}{k!}, \quad k > 0, \quad W_{jl} > 0 \tag{7}
$$

But this also holds for $W_{jl} \le 0$ since $|J_k(-u)| = |J_k(u)|$, and for $k \le 0$ as $|J_{-k}(u)| = |(-1)^k J_k(u)| = |J_k(u)|$. Then, substituting equation 7 in $\alpha_{\mathbf{k}} = \prod_{l=1}^n J_{k_l}(W_{jl})$, we obtain the desired result. $\square$

## A.2 THEOREM 2

**Theorem 4.** *Let $f_\theta$ be a sinusoidal INR of depth $d$, and let $\widetilde{f}_\theta$ be the network obtained by perturbing the $k$-th hidden layer weights and biases to $\widetilde{\mathbf{W}}^k$ and $\widetilde{\mathbf{b}}^k$. Then,*

$$
\sup_{\mathbf{x}} \left\| f_\theta(\mathbf{x}) - \widetilde{f}_\theta(\mathbf{x}) \right\|_\infty \le \left( \|\mathbf{W}^k - \widetilde{\mathbf{W}}^k\|_\infty + \|\mathbf{b}^k - \widetilde{\mathbf{b}}^k\|_\infty \right) \|\mathbf{L}\|_\infty \prod_{i=k+1}^d \|\mathbf{W}^i\|_\infty.
$$

*Proof.* First, note that $\mathbf{x} \mapsto \mathbf{L}\mathbf{x}$ is $\|\mathbf{L}\|_\infty$-Lipschitz for infinity norms:

$$
\|\mathbf{L}\mathbf{x} - \mathbf{L}\mathbf{x}'\|_\infty = \|\mathbf{L}(\mathbf{x} - \mathbf{x}')\|_\infty \le \|\mathbf{L}\|_\infty \|\mathbf{x} - \mathbf{x}'\|_\infty.
$$

Second, note that $\mathbf{x} \mapsto \sin(\mathbf{x})$ is 1-Lipschitz also for infinity norms, and thus $\mathbf{x} \mapsto \mathbf{S}^i(\mathbf{x}) = \sin(\mathbf{W}^i\mathbf{x} + \mathbf{b}^i)$ is also $\|\mathbf{W}^i\|_\infty$-Lipschitz:

$$\| \sin(\mathbf{W}^i\mathbf{x} + \mathbf{b}^i) - \sin(\mathbf{W}^i\mathbf{x}' + \mathbf{b}^i)\|_\infty \leq \| \sin \|_{\mathrm{Lip}} \|(\mathbf{W}^i\mathbf{x}' + \mathbf{b}^i) - (\mathbf{W}^i\mathbf{x}' + \mathbf{b}^i)\|_\infty$$
$$\leq \|\mathbf{W}^i(\mathbf{x} - \mathbf{x}')\|_\infty \leq \|\mathbf{W}^i\|_\infty \|\mathbf{x} - \mathbf{x}'\|_\infty.$$

It thus follows that, for any $\mathbf{x}$:

$$\left\| (\mathbf{L} \circ \mathbf{S}^d \circ \cdots \circ \mathbf{S}^k \circ \cdots \circ \mathbf{S}^0)(\mathbf{x}) - (\mathbf{L} \circ \mathbf{S}^d \circ \cdots \circ \widetilde{\mathbf{S}}^k \circ \cdots \circ \mathbf{S}^0)(\mathbf{x}) \right\|_\infty$$

$$\leq \|\mathbf{L}\|_\infty \left\| (\mathbf{S}^d \circ \cdots \circ \mathbf{S}^k \circ \cdots \circ \mathbf{S}^0)(x) - (\mathbf{S}^d \circ \cdots \circ \widetilde{\mathbf{S}}^k \circ \cdots \circ \mathbf{S}^0)(\mathbf{x}) \right\|_\infty$$

$$\leq \|\mathbf{L}\|_\infty \left( \prod_{i=k+1}^d \|\mathbf{W}^i\|_\infty \right) \left\| (\mathbf{S}^k \circ \cdots \circ \mathbf{S}^0)(\mathbf{x}) - (\widetilde{\mathbf{S}}^k \circ \cdots \circ \mathbf{S}^0)(\mathbf{x}) \right\|_\infty$$

$$= \|\mathbf{L}\|_\infty \left( \prod_{i=k+1}^d \|\mathbf{W}^i\|_\infty \right) \left\| \sin(\mathbf{W}^k(\mathbf{S}^{k-1} \circ \cdots \circ \mathbf{S}^0)(\mathbf{x}) + \mathbf{b}^k) \right.$$
$$\left. - \sin(\widetilde{\mathbf{W}}^k(\mathbf{S}^{k-1} \circ \cdots \circ \mathbf{S}^0)(\mathbf{x}) + \widetilde{\mathbf{b}}^k) \right\|_\infty$$

$$\leq \|\mathbf{L}\|_\infty \left( \prod_{i=k+1}^d \|\mathbf{W}^i\|_\infty \right) \| \sin \|_{\mathrm{Lip}} \left\| (\mathbf{W}^k(\mathbf{S}^{k-1} \circ \cdots \circ \mathbf{S}^0)(\mathbf{x}) + \mathbf{b}^k) \right.$$
$$\left. - (\widetilde{\mathbf{W}}^k(\mathbf{S}^{k-1} \circ \cdots \circ \mathbf{S}^0)(\mathbf{x}) + \widetilde{\mathbf{b}}^k) \right\|_\infty$$

$$= \|\mathbf{L}\|_\infty \left( \prod_{i=k+1}^d \|\mathbf{W}^i\|_\infty \right) \left\| (\mathbf{W}^k - \widetilde{\mathbf{W}}^k)(\mathbf{S}^{k-1} \circ \cdots \circ \mathbf{S}^0)(\mathbf{x}) + (\mathbf{b}^k - \widetilde{\mathbf{b}}^k) \right\|_\infty$$

$$\leq \|\mathbf{L}\|_\infty \left( \prod_{i=k+1}^d \|\mathbf{W}^i\|_\infty \right) \left( \left\| (\mathbf{W}^k - \widetilde{\mathbf{W}}^k)(\mathbf{S}^{k-1} \circ \cdots \circ \mathbf{S}^0)(\mathbf{x}) \right\|_\infty + \left\| \mathbf{b}^k - \widetilde{\mathbf{b}}^k \right\|_\infty \right)$$

$$\leq \|\mathbf{L}\|_\infty \left( \prod_{i=k+1}^d \|\mathbf{W}^i\|_\infty \right) \left( \|\mathbf{W}^k - \widetilde{\mathbf{W}}^k\|_\infty \left\| (\mathbf{S}^{k-1} \circ \cdots \circ \mathbf{S}^0)(\mathbf{x}) \right\|_\infty + \left\| \mathbf{b}^k - \widetilde{\mathbf{b}}^k \right\|_\infty \right).$$

Finally, note that since sines lie in $[-1, +1]$, it must hold that $\left\| (\mathbf{S}^{k-1} \circ \cdots \circ \mathbf{S}^0)(\mathbf{x}) \right\|_\infty = \max_i \left| [(\mathbf{S}^{k-1} \circ \cdots \circ \mathbf{S}^0)(\mathbf{x})]_i \right| \leq \max_i 1 = 1$, from which we conclude the proof. $\square$

## B  SIGNED DISTANCE FUNCTIONS

Table 5: Quantitative comparisons on representing surfaces from the Stanford 3D Scanning Repository with $\omega_0 = 30$. We compare the adapted network trained with \method{}, the large network with standard training (large), and the model with small architecture and standard training (small). We report the average chamfer distance (Avg CD ($\times 10^2$)) between reconstructed and ground-truth surfaces and the percentage of network parameters compared to the large architecture (lower is better).

| Model (SDFs) | Variant | CD ($\times 10^2$) ↓ | Size reduct. ↑ |
|---|---|---|---|
| SIREN | Large | **0.56 ± 0.08** | - |
| | Small | 0.59 ± 0.09 | 62.14 |
| | Ours | 0.58 ± 0.06 | 62.14 |
| FINER | Large | **0.63 ± 0.09** | - |
| | Small | 0.67 ± 0.09 | 62.14 |
| | Ours | **0.63 ± 0.06** | 62.14 |

In Table 5 we evaluate AIRe ('Ours') against an overparametrized, large INR of size $[256, 256, 256]$ and a small, reduced model of size $[128, 128, 256]$. These last two are fitted with the standard training pipeline, and the reconstruction quality was measured using the Chamfer Distance ($\times 10^2$).

Table 6 shows a per-scene breakdown of the SDF quantitative results presented in the main paper when $\omega_0 = 60$ and and the small network size is $[64, 64, 256]$. The per-scene breakdown is consistent with the aggregate quantitative metrics. Our method outperforms the small network in all cases. It also obtains similar or better accuracy compared to the large network but uses roughly 1/6 of network parameters.

Table 6: Per-scene quantitative comparisons on representing surfaces from the Stanford 3D Scanning Repository with $\omega_0 = 60$ and model size $[64, 64, 256]$. We compare AIRe, the network with large architecture, and the model with small architecture. We report the chamfer distance (CD ($\times 10^2$)) between reconstructed and ground-truth surfaces (lower is better). Best values in **bold**, second best values underlined.

| Model | Variant | CD ($\times 10^2$) $\downarrow$ | | | | |
|---|---|---|---|---|---|---|
| | | **Armadillo** | **Bunny** | **Dragon** | **Happy Buddha** | **Lucy** |
| SIREN | Large | **0.60** | 0.75 | 0.65 | **0.50** | 0.74 |
| | Small | 0.99 | 0.79 | 0.82 | 0.98 | 0.86 |
| | Ours | 0.65 | **0.69** | **0.62** | 0.64 | **0.61** |
| FINER | Large | 2.13 | 2.06 | 2.17 | 2.74 | 1.60 |
| | Small | 5.51 | 10.80 | 4.57 | 2.58 | 1.92 |
| | Ours | **0.88** | **0.95** | **0.73** | **1.10** | **0.76** |

Figure 7 shows additional examples of surface reconstructions using SIREN with $\omega_0 = 60$ and model architecture $[64, 64, 256]$. As in the other examples, the surface trained using our method presented a lower error compared to the small network. We also see in Figure 8 an example using FINER with settings $\omega_0 = 60$ and network architecture $[64, 64, 256]$. Note that AIRe offers a better reconstruction than the small network with less artifacts.

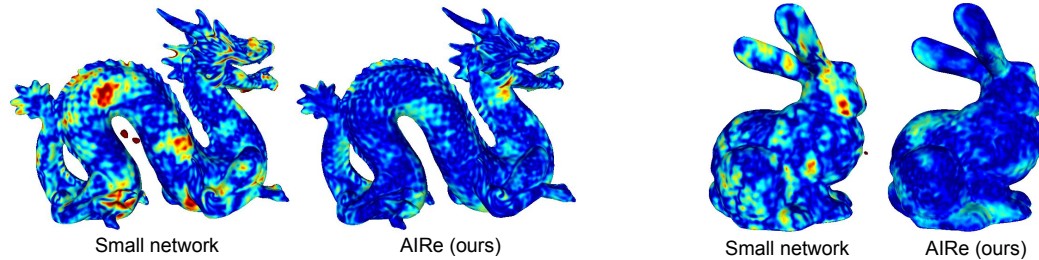

Figure 7: Additional qualitative comparisons on representing surfaces based on the Dragon and Bunny models using a SIREN network with $\omega_0 = 60$ and model architecture $[64, 64, 256]$. Left: results of training the small network. Right: results of AIRe. We illustrate the unsigned distance from the ground-truth surface using a color scale from dark blue (zero) to dark red ($\geq 0.01$).

We also present Table 7, where we compare the time overhead when training the large, small and adapted models,. Observe that the AIRe and the large model have an equivalent training time but with only $16.04\%$ of the original parameters. Furthermore, observe that a small model trained during the same amount of time has much worse accuracy than a network trained with AIRe.

**Neural SDF evolution.** Finally, we evaluated AIRe on a surface-smoothing task governed by the mean curvature equation, a classical PDE widely used in geometry processing (Mehta et al., 2022; Yang et al., 2021). In this setting, the network parametrizes a 4D solution $f : \mathbb{R}^4 \to \mathbb{R}$ of

$$\frac{\partial f}{\partial t} = \alpha \, \mathrm{div}\left( \frac{\nabla_{\mathbf{x}} f}{\|\nabla_{\mathbf{x}} f\|} \right), \quad \text{subject to} \quad f(\mathbf{x}, 0) = g(\mathbf{x}),$$

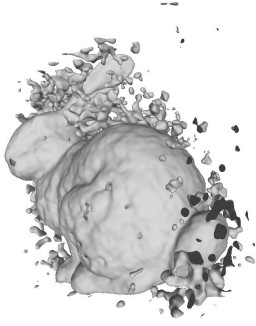 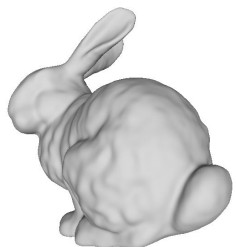

Small network                                          AIRe (ours)

Figure 8: A case with the Bunny model using a FINER network with $\omega_0 = 60$ and model size $[64, 64, 256]$. The final, small network (left) presents several artifacts, which does not occur with the AIRe method (right).

Table 7: **Time overhead comparisons when training SDFs.** We consider a *Large* network with $\sim 133K$ parameters and a *Small* network with $\sim 22K$ parameters that are fitted for 1000 epochs. We compare them with a network adapted during 1000 epochs using AIRe up to a size equal to the small network. We also train the final network for 2000 epochs to compare the reconstruction quality along a time budget.

| Variant | CD ($\times 10^2$) ↓ | Size reduct. ↑ | Time (s) ↓ |
|---|---|---|---|
| Large | 0.65 | - | 76.0 |
| Small ($10^3$ ep) | 0.89 | 83.96% | 39.4 |
| Small ($2\times 10^3$ ep) | 0.86 | 83.96% | 83.2 |
| Ours | 0.64 | 83.96% | 76.8 |

| SDF | Large | Small ($10^3$ ep) | Small ($2\times 10^3$ ep) | Ours |
|---|---|---|---|---|
| Armadillo | 52 | 27 | 55 | 52 |
| Bunny | 34 | 17 | 35 | 34 |
| Dragon | 60 | 31 | 65 | 60 |
| Happy Buddha | 159 | 83 | 179 | 162 |
| Lucy | 75 | 39 | 82 | 76 |
| Avg. time (s)↓ | 76.0 | 39.4 | 83.2 | 76.8 |

where $g$ is the SDF of the initial surface, $\nabla_{\mathbf{x}} f$ is the spatial gradient, $\mathrm{div}$ denotes the divergence operator, and $\alpha$ controls the smoothing rate. We adapted AIRe within the NISE framework (Novello et al., 2023) to solve this PDE using the Armadillo SDF as the initial condition, over the time interval $t \in [0, 0.2]$, with $\alpha = 0.001$. The network $f$ was trained during 10000 epochs using an oriented point cloud of size $\approx 173000$. During training, we used minibatches of 15000 on-surface points, 15000 off-surface points, and 30000 in $\mathbb{R}^3 \times [-1, 1]$.

We begin with an architecture of $[256, 256, 256]$ and prune approximately 34% of its parameters by removing 25% of the neurons from the first two layers. We then compare the resulting evolution against a baseline obtained by training the original (larger) network from scratch. Since no ground-truth evolution is available for this task, we evaluate the result by reconstructing the evolved surfaces at $t = 0.0, 0.1, 0.2$ for both models and computing the CD between them, obtaining an average error of 0.004. This indicates that AIRe adapts the network while preserving the accuracy of the geometric evolution. Figure 9 shows the reconstructed surfaces at these three time steps; as expected, regions of high curvature contract as the surface smooths over time.

## C  IMAGES

In Figure 10, we evaluate the effect of our densification and pruning schemes on both training convergence and spectral stability. Our training procedure begins with an initial fitting stage, followed by a densification step in which newly added input neurons are initialized with small contributions (via low column norms) to prevent a spike in the loss. The model is then fine-tuned (blue region in the training curve), which produces a sharp drop in training error and increases the contributions of the added neurons (see Figure 10, bottom). As a result, most newly added neurons are no longer classified as low-contribution (orange points) by the time TWD begins.

After selecting candidate neurons for pruning, we apply TWD to progressively reduce their influence on the output. This regularization introduces a small bump in the loss near epoch 2400, but the curve

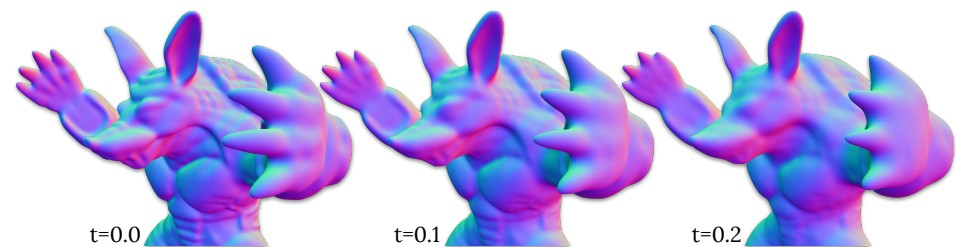

t=0.0      t=0.1      t=0.2

Figure 9: **Surface smoothing via mean curvature flow using AIRe.** Reconstructed time steps ($t = 0.0$, $t = 0.1$, $t = 0.2$) of the Armadillo model evolving under the mean curvature equation. Despite pruning approximately 34% of the parameters, AIRe accurately captures the geometric evolution, achieving a final CD of 0.004 with respect to the baseline large model.

quickly stabilizes and remains below the loss achieved by the large network. Furthermore, TWD mitigates the discontinuity normally caused by pruning, as it minimizes the discrepancy between the reconstructions immediately before and after the pruning step. This effect is illustrated in Figure 10 (top-right), which shows the difference between the Fourier transforms of the reconstructed signal before and after pruning: pruning without TWD produces significant spectral artifacts, whereas pruning with TWD preserves the spectrum much more faithfully.

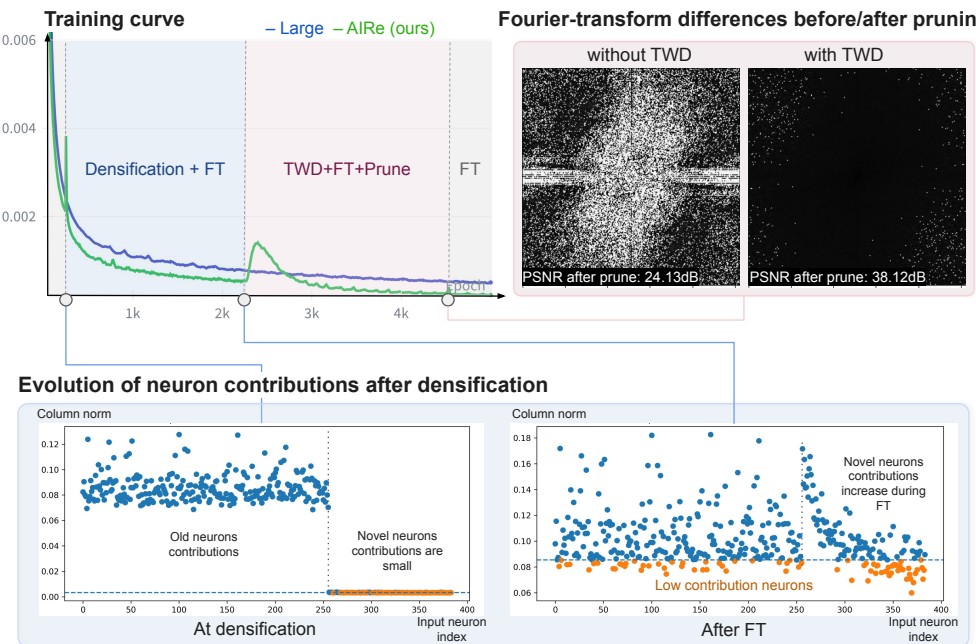

Figure 10: **Effect of densification and pruning on convergence and spectral stability.** *Top left:* Training curve showing the stages of AIRe (initial fitting; densification; fine-tuning; TWD followed by pruning). *Top right:* Fourier-transform differences between reconstructed signals before and after pruning, comparing pruning without TWD and with TWD. Pruning without TWD leads to strong spectral distortion (PSNR 24.13dB), whereas pruning with TWD preserves the spectrum (PSNR 38.12dB). *Bottom:* Evolution of neuron contributions after densification. Left: at densification time, newly added neurons have small norms. Right: after fine-tuning, the contributions of novel neurons increase, while low-contribution neurons (orange) become clear pruning candidates.

Table 8: **AIRe applied to SPDER architecture.**

| Model (Images) | Variant | PSNR↑ | Size reduct.↑ |
|---|---|---|---|
| SPDER | Large | 30.77 | – |
| SPDER | Ours | 34.61 | 34.89% |

**Performance evaluation.** Table 8 evaluates AIRe on the SPDER architecture in the DIV2K dataset, using the same model size and training configuration as in Table 1. We use 90% of the pixels for training and 10% for computing PSNR. As shown in the table, AIRe improves reconstruction quality by nearly 4dB while reducing the number of parameters by 34.89%, demonstrating its ability to effectively adapt distinct INR architectures to a given signal.

Table 9: **Inference performance for the image task.**

| Variant | CPU inf. time (s)↓ | GPU inf. time ($\times 10^{-4}$ s)↓ | GPU avg usage (%)↓ |
|---|---|---|---|
| Large | 0.74 | 4.32 | 44.17 |
| Ours | 0.61 | 3.19 | 31.53 |
| Improvement | 17.16% | 26.17% | 28.62% |

Table 9 reports the CPU/GPU inference time and the average GPU usage for the image models trained in Table 3. Each measurement was repeated four times per network to ensure robustness. Our method achieves a reduction of 17.16% in CPU inference time and 26.17% in GPU inference time, while also lowering average GPU utilization by 28.62%. This reduction in resource usage enables larger image batches to be processed during inference.

**Additional ablations.** We present ablation studies to support the choice of hyperparameters for AIRe. First, we investigate the optimal allocation of epochs between the targeted weight decay stage and the fine-tuning stage under a fixed training budget. Specifically, we train SIREN Sitzmann et al. (2020) and FINER Liu et al. (2024) models, each with two hidden layers of 512 neurons, for a total of 5000 epochs. Training begins with standard optimization for $x$ epochs, followed by targeted weight decay for $y$ epochs, where $x, y \in \{100, 750, 1000, 1250, 1500, 1750, 2000, 2250\}$. The remaining $5000 - x - y$ epochs are allocated to fine-tuning.

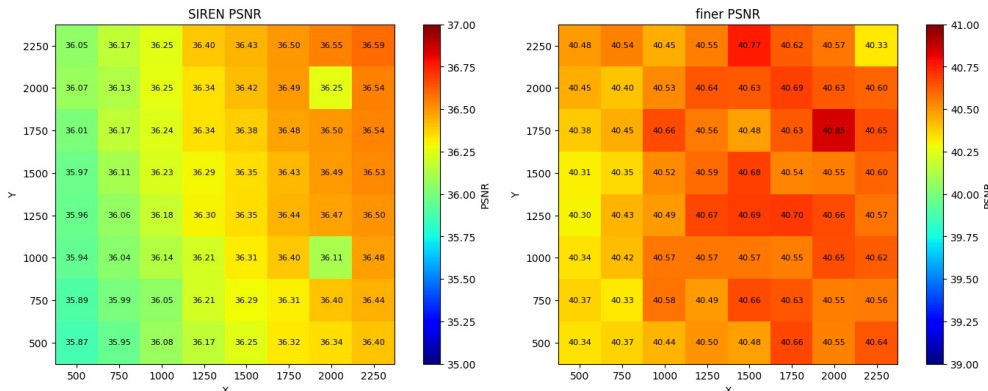

Figure 11: Ablation of the number of epochs used to pre-train the model (x axis), to train with targeted weight decay (y axis) and fine tune. The total training lasts for 5000 epochs, and each value refers to the mean PSNR over the DIV2K dataset. (Left) SIREN Sitzmann et al. (2020) architecture shows that longer standard training and targeted weight decay stage improve quality, even with fewer fine tuning epochs. (Right) FINER architecture shows less consistency in the results, demonstrating that above 1000 epochs of standard training the results improve, but show no clear pattern.

Figure 11 shows the PSNR for each epoch distribution, where the $x$-axis corresponds to the number of epochs used for the initial standard training stage, and the $y$-axis indicates the number of epochs

allocated to the targeted weight decay stage. As shown, SIREN models benefit from increased training time in both the standard training and targeted weight decay stages, resulting in improved reconstruction accuracy. In contrast, FINER models show only marginal improvements when the initial training stage exceeds 1000 epochs.

To determine the optimal pruning configuration, both in terms of which layers to prune and the amount per layer, we train models with the best-performing epoch distribution for both SIREN and FINER over 5000 epochs, applying varying levels of pruning to each layer. Figure 12 presents the PSNR of each reconstruction, where $x$ represents the percentage of neurons pruned in the first layer, and $y$ denotes the percentage pruned in the second layer.

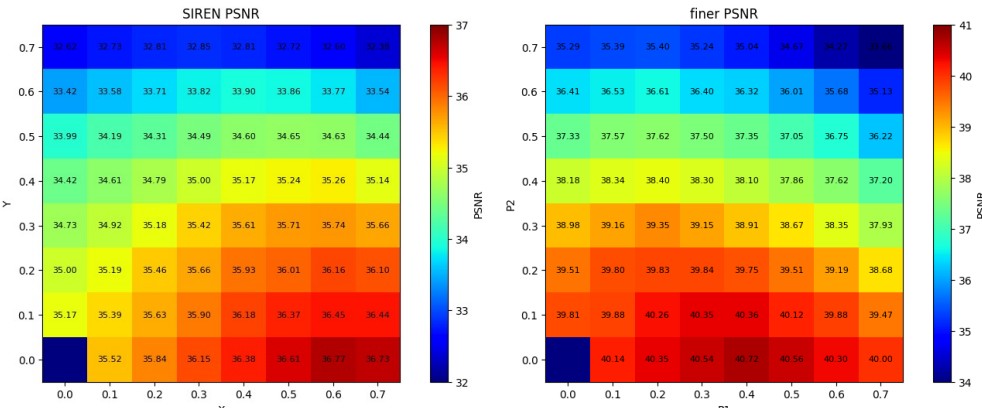

Figure 12: Ablation of the quality degradation with respect to the percentage of prune on the input neurons (x axis) and the 1st hidden neurons (y axis). The total training lasts for 5000 epochs, and each value refers to the mean PSNR over the DIV2K dataset. Observe that both images show that pruning the 1st layer retains more quality than pruning the 2nd layer. (Left) A SIREN Sitzmann et al. (2020) architecture reconstruction quality is preserved even with an extreme prune of 60%. (Right) FINER architecture quality is better retained when pruning the first layer, albeit with less percentage.

Both SIREN and FINER benefit from pruning the input neurons, although the optimal percentage of neuron removal differs between the two. In contrast, pruning hidden neurons generally leads to a degradation in reconstruction quality.

We also perform an ablation study on the use of regularization to improve neuron removal during training. Specifically, we train a sinusoidal INR using three configurations: standard weight decay, targeted weight decay, and no regularization prior to pruning. Standard weight decay yields the lowest reconstruction accuracy at 34.2dB, while removing regularization improves the result by 0.51dB. The targeted weight decay stage achieves the best performance, increasing accuracy to 36.9 dB.

For the densification strategy, we examine the impact of varying both the number of training epochs before densification and the percentage of new input neurons added. Concretely, the INR is initially trained for $x$ epochs, then its first layer is expanded by $(y * 100)\%$, and the augmented network is fine-tuned for the remaining $3000 - x$ epochs. The results are presented in Figure 13.

As expected, increasing the number of neurons leads to higher PSNR values. Additionally, accuracy improves when a larger number of neurons is added early in the training process (i.e., before 1500 epochs).

## D    ADDITIONAL DISCUSSIONS

We provide further details regarding the parameter settings used in our experiments. The targeted weight decay stage is trained using the following loss function

$$\mathcal{L}_{\alpha,\mathcal{I}} = \mathcal{L}_{\text{data}} + \alpha \sum_{j \in \mathcal{I}} \|\mathbf{W}_{*j}\|_1,$$

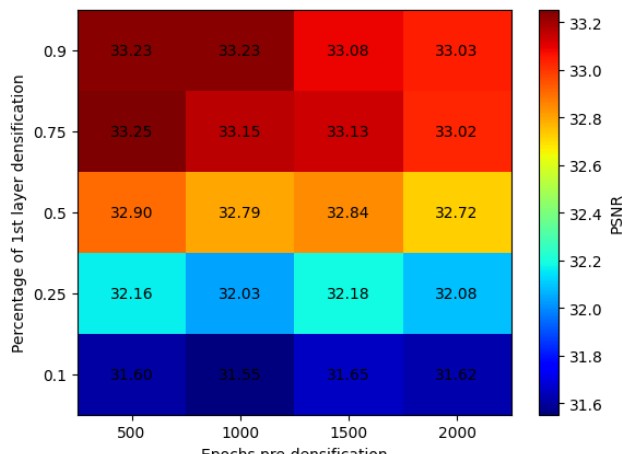

Figure 13: Ablation on the epochs trained before densifying ($x$ axis) compared to the percentage of input neurons added ($y$ axis).

where $\alpha$ is a parameter that starts at zero and increases linearly up to one at the end of this stage. For the pruning scheme, we use the Prune package in PyTorch, using structured masks over the to remove the corresponding weights. Specifically, when pruning neuron $h_j^i$, we mask the entries of the $j$-th column of $\mathbf{W}^i$. This removes all the neuron's influence from the network. As for the densification technique, we preserve the optimizer state of previous neurons to minimize training affectation.

We trained using a 12 GB NVIDIA GPU (TITAN X Pascal) and a 24 GB NVIDIA GPU (RTX 4090).

### D.1 NEURAL RADIANCE FIELDS

We evaluate the forward-pass latency, computational cost (GFLOPs) for the large SIREN model and our pruned architecture (AIRe) on the trained models in Table 1. The results are summarized in Table 10.

These results show that computational cost is reduced by $\approx 31\%$, consistent with the structural sparsity introduced by our pruning mechanism. This indicates that removing neurons and reducing FLOPs directly accelerates the forward pass on GPU. Overall, these measurements confirm that our architectural adaptation yields meaningful improvements in GPU inference latency, which is the critical metric for real-time or high-resolution INR workloads.

Table 10: **Inference efficiency comparison.**

| Variant | Gflops |
|---------|--------|
| Large   | 7.49   |
| AIRe    | 5.17   |
| Improv. | 30.96% |

We adopt the torch-ngp framework [2] for NeRF implemented by FINER that considers two networks: A density network that takes a 3D position as input and outputs a density value and a geometric feature vector $v \in \mathbb{R}^{182}$; and a color network that receives $v$ and a 3D direction and returns a RGB color. NeRF computes the color of a pixel with volume rendering using 3D points sampled on a ray traced from the center of the virtual camera through the pixel Mildenhall et al. (2021). We set the batch size to $4,096$ rays and Adam optimizer with learning rate of $0.0002$, $\beta_1 = 0.9$, $\beta_1 = 0.99$, $\epsilon = 10^{-15}$, and exponential learning rate decay of $0.1$. We update the model weights using an exponential moving average with a decay of $0.95$. We follow FINER's experimental setting, where for each scene of the Blender dataset, we have 25 images for training, 200 images for testing, all downsampled to $200 \times 200$ pixels. We employ the PSNR and the number of network parameters (Params) as evaluation metrics.

We evaluate three approaches for NeRF training: **Large** and **Small** networks, which train from scratch for $1.5 \times 10^3$ epochs a density network of size $[182, 182, 182]$ and color networks with architecture $[182, 182, 182]$ and $[91, 91, 182]$, respectively. On the other hand, **AIRe** considers training from scratch for 300 epochs the same networks from Large model, then selecting $50\%$ of the input neurons

---

[2]https://github.com/ashawkey/torch-ngp

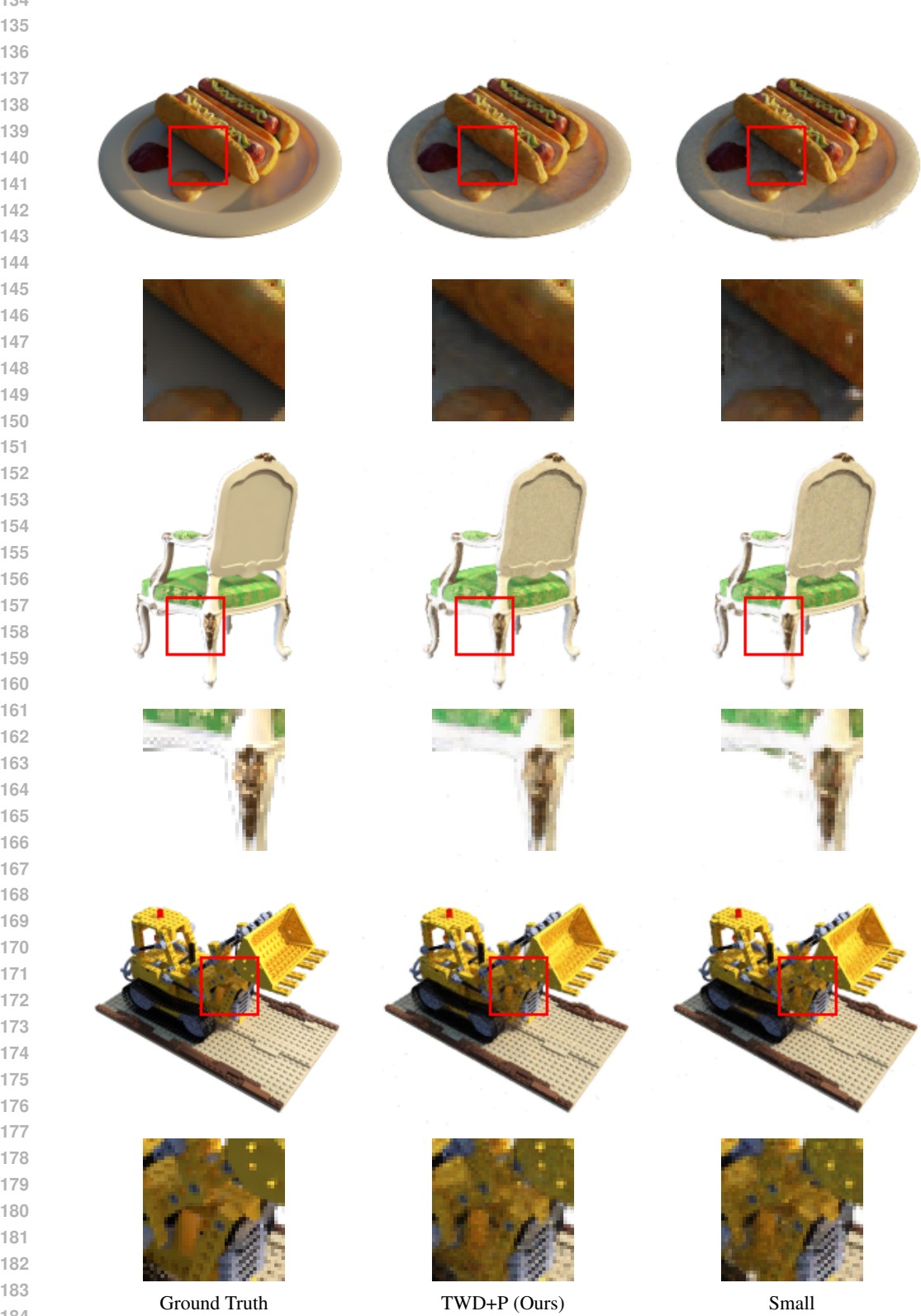

Ground Truth · TWD+P (Ours) · Small

Figure 14: Qualitative comparisons on representing NeRFs using the Hotdog (top), Chair (middle), and Lego (bottom) scenes from the Blender dataset. First column: ground-truth views. Second column: results of our proposed approach of targeted weight decay for pruning (TWD+P). Third column: results of training from scratch a small network (Small) with an architecture equivalent to ours after pruning. Differences in quality highlighted by insets.

and the first hidden neurons of the density network for 750 epochs of TWD, followed by pruning of selected neurons, and finally 450 epochs of fine-tuning of both density and color networks.

Table 11: Quantitative comparisons between AIRe's pruning scheme with training from scratch the 'Large' and 'Small' networks. We report the average PSNR between reconstructed and ground-truth test views (higher is better), the PSNR difference with respect to Large (higher is better), and the percentage of network parameters with respect to Large (higher is better). Best values in **bold**, second best values underlined.

| | Method | Chair | Drums | Ficus | Hotdog | Lego | Materials | Mic | Ship | Avg | Size reduct. |
|---|---|---|---|---|---|---|---|---|---|---|---|
| PSNR← | Large | **34.04** | **24.81** | **28.84** | **33.42** | **29.96** | **27.01** | **33.96** | **22.55** | **29.32** | - |
| | Small | 33.12 | 24.14 | 27.77 | 32.06 | 28.75 | 26.47 | 33.68 | 22.28 | 28.53 | **20.74%** |
| | Ours | 33.23 | 24.11 | 27.82 | 33.10 | 28.82 | 26.21 | 33.59 | 22.26 | 28.64 | **20.74%** |

Table 11 shows that compared to Large, the decrease in PSNR of AIRe was $13.9\%$ lower than the PSNR of the Small network approach, even when both have the same number of network parameters. The pruning procedure allows our NeRF to save more than $20\%$ of network parameters compared to the Large approach. We also see qualitative improvements compared to the Small network, such as shadows/bright spots in the Hotdog (see Figure 1).

