# OpenReview forum: "Adaptive Training of INRs via Pruning and Densification"
_ICLR.cc/2026/Conference — Submitted to ICLR 2026_

### Official Review · Reviewer_aRcV · 2025-10-23

**Soundness:** 3
**Presentation:** 3
**Contribution:** 3
**Rating:** 4
**Confidence:** 4

**Summary:**

This paper introduces AIRe (Adaptive Implicit neural Representation), a novel adaptive training framework designed to refine the architecture of Implicit Neural Representations (INRs), specifically those based on sinusoidal functions, during the optimization process. The primary goal is to address the challenge of parameter redundancy.

**Strengths:**

1. The paper introduces AIRe, a novel adaptive training framework that effectively combines neuron densification and pruning
2. The numerical results consistently confirm the theoretical assumptions laid out for the AIRe framework

**Weaknesses:**

1. Lack of comparison with:
- SPDER: Semiperiodic Damping-Enabled Object Representation
- FreSh: Frequency Shifting for Accelerated Neural Representation Learning

2. Authors do not commenton the  relation between pruning and densification in INR and Gaussian Splatting
- 3D Gaussian Splatting for Real-Time Radiance Field Rendering

which can be interesting for readers.

3. In related works, authors should mention that Gaussian Components can also reconstruct 2D images

- GaussianImage: 1000 FPS Image Representation and Compression by 2D Gaussian Splatting
- MiraGe: Editable 2D Images using Gaussian Splatting

4. Since the method introduces multiple additional training phases (densification, TWD, fine-tuning), the real efficiency gain remains unclear.

5. The experimental evaluation primarily contrasts AIRe with generic pruning algorithms such as RigL and DepGraph, which are not tailored to INRs.

6. The proposed densification and pruning schedules rely on manually chosen hyperparameters (e.g., number of added neurons, pruning thresholds, or training epochs).

7. The ablation studies show that the densification component offers little benefit for the FINER architecture.

**Questions:**

1. How does AIRe compare quantitatively with recent adaptive INR methods such as SPDER or FreSh?

2. Could the authors clarify the conceptual relation between pruning/densification in INRs and Gaussian Splatting techniques?

3. Why are recent 2D Gaussian-based representations (e.g., GaussianImage, MiraGe) not discussed in Related Work?

4. What are the actual training and inference time gains achieved by AIRe after including the extra adaptation phases?

5. Why are only generic pruning baselines (RigL, DepGraph) considered instead of INR-specific adaptive models?

6. Are the pruning and densification hyperparameters selected manually or adaptively during training?

7. Why does the densification component provide little or no improvement for the FINER architecture?

---

> ### Author Response · Authors · 2025-11-26
>
> We are thankful for the thoughtful comments. It is encouraging that the reviewer found our framework novel and our results aligned to the theoretical assumptions. We address their comments below.
>
> **Comparison with SPDER.**
>
> We tested AIRe’s adaptation on SPDER [1] with the same training configuration as in Table 3. We followed the evaluation protocol described in our paper, considering 90% of the pixels for training and 10% for evaluating PSNR. The results are presented in Table 8:
>
> _Table 8: AIRe applied to SPDER architecture._
> |**Model (Images)**|**Variant**|**PSNR**|**Size reduct.**|
> |-|-|-|-|
> |SPDER|Large|30.77|-|
> |SPDER|Ours|34.61|34.89% |
>
> Our training improves reconstruction quality (~4dB) with less parameters (34.89% model reduction) compared to the large (baseline) network. This experiment illustrates the potential of AIRe to adapt the network architecture and even improve generalization on INR architectures.
>
> Regarding the reviewer’s suggestion to include FreSh [2], we are currently running this experiment and expect to present it in the next few days. Nevertheless, FreSh operates directly on the standard SIREN architecture, and our method (AIRe) is designed precisely to enhance and adapt such INRs. For this reason, we expect AIRe to be competitive in this setting as well.
>
> **Pruning and densification terms in INRs vs. Gaussian Splatting.**
>
> The terms densification and pruning is inspired by analogous concepts in Gaussian Splatting [3]. However, the underlying mechanisms are fundamentally different: in our case, these operations correspond to the addition or removal of neurons within an INR, whereas in Gaussian Splatting they involve adding or removing explicit 3D splats used to model a scene. We clarified this and cited the Gaussian Splatting works relevant to image reconstruction [4, 5] in the paper (L126-L128).
>
> **Efficiency gain during training.**
>
> Figures 1 (right), 4, and 8 show that our dynamic adaptation scheme mitigates training divergence due to overparametrization of the network. Besides, our method has better accuracy across a time budget in the SDF fitting task, as shown in Table 7 (left).
>
> _Table 7: Time overhead comparisons when training SDFs._
> |**Variant**|**CD (×10²)**|**Size reduct.**|**Time (s)**|
> |-|-|-|-|
> |Large|0.65|-|76.0|
> |Small (10³ ep)|0.89|16.04%|39.4|
> |Small (2×10³ ep)|0.86|16.04%|83.2|
> |Ours|0.64|16.04%|76.8|
>
> Our method fits a signal with comparable quality to the large model, but with fewer parameters; this translates to significant gains during inference, including faster CPU/GPU inference times, as well as lower GPU usage (see Table 9).
>
> _Table 9: Inference performance for the image task._
> ||**CPU inference time (s)**|**GPU avg usage (%)**|**GPU inference time (x10^-4s)**|
> |-|-|-|-|
> |Large|0.74|44.17|4.32|
> |Ours|0.61|31.53|3.19|
> |Improvement|17.16%|28.62%|26.17%|
>
> **Comparisons with generic pruning methods.**
>
> We agree with the reviewer that the pruning methods we compare to are not specifically adapted to INRs.
> We have now implemented Zell et al. [6] INR pruning scheme independently and added a comparison to their method using the DIV2K dataset in Table 2, as a more appropriate benchmark.
>
> _Table 2 (Partial): Comparison of AIRe with Zell et al. [6]_
> |**Method**|**PSNR**|**SSIM**|
> |-|-|-|
> |Zell et al.|18.96±1.92|0.39±0.08|
> |AIRe|37.53±3.75|0.96±0.02|
>
> **Choice of hyperparameters.**
>
> To obtain a fair comparison between methods, most of our experiments indeed consider a fixed number of added or pruned neurons. However, note that the number of pruned neurons across layers in Table 1 is automatically chosen by our method. Moreover, we also evaluate the change in PSNR over varying percentages of pruned neurons per layer in Figure 12, and over varying percentages of densified neurons in Figure 13.
>
> **Densification in the FINER architecture.**
>
> We introduce a densification scheme to increase the network spectrum coverage at training. As shown in SIREN’s column in Table 4 and the training curve in Figure 10, this is particularly important when the signal has many details and the network has not enough parameters. On the other hand, the FINER architecture was designed to enhance the spectrum content, thus reducing the impact of densification. Instead, it greatly benefits from pruning, especially when representing SDFs (Table 3).
>
> [1] Shah, K., and Chawin,S.. *SPDER: Semiperiodic Damping-Enabled Object Representation.* ICLR 2024
>
> [2] Kania, A., et al. *FreSh: Frequency Shifting for Accelerated Neural Representation Learning.* ICLR 2025
>
> [3] Kerbl, B., et al. *3D Gaussian splatting for real-time radiance field rendering.* SIGGRAPH 2023
>
> [4] Zhang, X., et al. *Gaussianimage: 1000 fps image representation and compression by 2d gaussian splatting.* ECCV 2024
>
> [5] Waczynska, J., et al. *MiraGe: Editable 2D Images using Gaussian Splatting.* ICML 2025
>
> [6] Benbarka, N., Timon H., and Andreas Z. *Seeing implicit neural representations as fourier series.* WACV 2022

---

> > ### Comment · Reviewer_aRcV · 2025-11-28
> >
> > The authors have adequately addressed all of my concerns in the rebuttal. In particular, the clarifications regarding the methodology, experimental protocol, and the additional evidence provided resolved my earlier doubts about correctness and reproducibility.
> > I raise the score.

---

### Official Review · Reviewer_HQzk · 2025-10-27

**Soundness:** 3
**Presentation:** 2
**Contribution:** 3
**Rating:** 4
**Confidence:** 4

**Summary:**

This paper proposes AIRe (Adaptive Implicit Neural Representation), an adaptive training framework for INR, which primarily employs neuron pruning and densification strategies to adjust the network. The introduction of the TWD mechanism allows information from low-contributing neurons to be transferred to important neurons, theoretically ensuring the stability of pruning. Input frequency densification enhances the network’s representational capacity. Experiments across multiple tasks demonstrate that AIRe can reduce model size while maintaining or even improving reconstruction quality.

**Strengths:**

1. The paper proposes pruning and densification strategies for INR, including the TWD mechanism to transfer information from low-contributing neurons.

2. Experiments across multiple tasks comprehensively validate the approach, showing that it reduces model size while maintaining or sometimes improving reconstruction quality.

3. The work has potential significance for pruning and densification in the INR domain.

**Weaknesses:**

1.Some theoretical explanations are unclear. It is not specified how the 2ωj frequency is determined during densification and why this particular frequency is chosen.

2.Discussion of pruning effects on input and hidden layers is limited.For the SDF task (Lines 324–334), hidden layers are pruned, while for the image fitting task (Lines 378–385), input neurons are pruned. The paper only mentions that pruning the input layer may harm reconstruction.

3.The pruning threshold ϵ is not clearly defined; it is unclear whether it is fixed, layer-wise adaptive, or percentile-based. Clarification on how ϵ is chosen or tuned would be helpful.

4.Experiments mainly report final PSNR or CD. Including spectral visualizations and convergence curves would better illustrate how pruning and densification affect optimization and frequency coverage.

5.Experimental setup is somewhat unclear. In Table 1, it is not specified whether the reported “large” baseline is based on SIREN, FINER, or something else.

6.Some figures (3, 4, 6) could be improved for clarity. Figures 3 and 4 could be redesigned to provide richer visual comparisons, and the right half of Figure 6 is somewhat confusing in the information it intends to convey.

**Questions:**

1.Why is densification performed using 2ωj for new neurons rather than other frequency choices? Is there a theoretical justification or is it empirically determined?

2.In the densify-before-prune schedule, are newly added neurons immediately considered low-contributing and pruned by TWD?

3.What does the “large” baseline in Table 1 correspond to (SIREN, FINER, or their average)?

4.What criteria guide the choice of pruning input versus hidden layers, and can more theoretical or experimental analysis be provided on the pruning effects on input and hidden layers?

---

> ### Author Response · Authors · 2025-11-26
>
> We thank the reviewer for the valuable feedback. We are encouraged to know they find potential in our work and that they consider that our experiments comprehensively validate our approach.
>
> **Chosen frequencies and new frequencies initialized during initialization.**
>
> We employ Theorem 1 to suggest natural frequencies candidates for densification:
> Multiples of input frequencies whose corresponding columns in the first hidden layer have high norm.
> Specifically, if the $j$th column $W^1_{*j}$ of the first hidden layer matrix $\mathbf{W}^1$ has a high norm, then the lowest multiples of its associated frequency $\omega_j$ tend to appear with larger amplitudes. This follows from Equation 3 and the fact that $|J_2(W^1_{ij})| > |J_{k_j}(W^1_{ij})|$ for most $i$, since Bessel functions are sorted by their order [1].
> Therefore, during densification, we identify input frequencies $\omega_j$ whose corresponding columns $W^1_{*j}$ exhibit high norms, and we expand the set of input frequencies $\omega$ by adding their doubled counterparts $2\omega_j$, boosting the input frequencies and thus enriching the network's representation.
> As shown in Figure 10, after densification most of the newly added neurons become highly contributive to the reconstruction.
>
> **Pruning of input and hidden neurons.**
>
> Actually, we perform pruning on both input neurons and hidden neurons for SDF, image, and NeRF representation in Table 1. We also have some experiments for the image fitting task where both input neurons and hidden neurons are pruned (Figure 5 and Figure 12). We have updated the paper to be more consistent regarding the input/hidden neurons nomenclature, which we hope made this more clear.
>
> **Choice of the pruning threshold ϵ.**
>
> The pruning threshold is set to $\epsilon=0.01$; it is fixed and applied uniformly across all layers. This prevents neurons with contributions above $\epsilon$ from being pruned, reducing the introduction of inaccuracies to the reconstruction. Specifically, note from Theorem 2 that $\epsilon$ bounds the term $\lVert W^k-\widetilde{W}^k \rVert_\infty+\lVert b^k-\tilde{b}^k \rVert_\infty$ and thus $\lVert f(x) - \tilde{f}(x) \rVert_\infty$ (as the remaining norms are controlled, typically $\leq1$), mitigating the effects of pruning. We have added this explanation to the start of the experiments section.
>
> **Analyzing densification and pruning via training curve and Fourier transform.**
>
> We have added Figure 10 to better illustrate the effects of our scheme on training convergence and reconstruction spectrum. In particular, we also present two subfigures showing the evolution of the densified input neuron contributions (according to column norm). There, we observe that after fine tuning and at the start of the TWD stage most of novel neurons have large contributions, and they are not selected to be pruned (i.e., they are not orange points).
>
> Moreover, note that Figure 1 illustrates a case in which the standard training of a large INR for SDF fitting diverges (due to network overparametrization), while AIRe stabilizes training.
>
>
> **Experimental setup of Table 1.**
>
> We clarified in the paper L273 that the models in Table 1 are defined with a SIREN architecture.
>
> **Improvement of figures.**
>
> Thank you for the feedback. We have slightly modified Figures 3 and 6 in an attempt to make them more clear. But overall we are unsure of what exactly the reviewer found confusing (especially in Figure 4); we'd appreciate it if they could specify it or mention what improvements could be made.
>
>
> [1] Novello, Tiago. *Understanding sinusoidal neural networks.* arXiv preprint arXiv:2212.01833 (2022).

---

> > ### Comment · Reviewer_HQzk · 2025-11-28
> > **Updated comments**
> >
> > Thanks for the authors’ efforts and clarifications, which resolved part of my concerns. I would like to raise my rating to borderline accept.

---

### Official Review · Reviewer_Bn28 · 2025-11-01

**Soundness:** 2
**Presentation:** 3
**Contribution:** 2
**Rating:** 6
**Confidence:** 3

**Summary:**

This paper introduces AIRe (Adaptive Implicit Neural Representation), a training framework that progressively adapts a potentially overparameterized INR to the target data through two complementary operations: pruning and densification of neurons. It provides a general
framework for the adaptive training of INRs, driven by pruning and densification. In the theoretical side, it leverage a harmonic expansion of sinusoidal neural networks to derive principled densification schemes, and prove stability of our neural networks under magnitudebased pruning. The method was mainly applied to SIREN and FINER for the experiments. Experiments were conducted on images, SDFs, and NeRFs,

**Strengths:**

The integration of pruning and frequency densification within INR training is innovative and addresses a key limitation—manual architecture tuning. In addition, the paper provides mathematical proofs (Theorem 1 and 2) explaining spectral densification and pruning stability, enhancing methodological rigor.

**Weaknesses:**

[1] The method only tested on low-dimensional signals (2D images, SDFs, small NeRF scenes). Therefore, it should be tested on different kinds of datasets. For example,  PDEs.

[2] One of the major drawbacks of INR is the long training time. By adding pruning and densification, will it increase the training time? An analysis of training time should be provided.

[3[How about the GPU comsumption? Like the Gfloop

[4]I understand that it was applied to SIREN and FINER and reports some results. However, some other baseline should also be added to the experiments.  For example, LosslessINR [r1]
Han, Woo Kyoung, et al. "Towards Lossless Implicit Neural Representation via Bit Plane Decomposition." Proceedings of the Computer Vision and Pattern Recognition Conference. 2025.

**Questions:**

See the weakness

---

> ### Author Response · Authors · 2025-11-26
>
> We appreciate the reviewer’s insightful comments. We are pleased to know they recognize the importance of the problem we tackle, considering our method as innovative and methodologically rigorous. Below, we address their main concerns.
>
> **PDEs & higher dimensions.**
>
> Indeed, our method is primarily designed for implicit neural representations (INRs) of low-dimensional signals in computer graphics and vision. However, we agree that evaluating AIRe on PDE-related tasks is an interesting and valuable direction.
>
> To explore this, we tested AIRe on a surface-smoothing task governed by the mean curvature equation, a well-known PDE in geometry processing. In this setting, the MLP represents a 4D function, as time is included as an input. We adapted AIRe to the NISE framework [1] and presented the full experiment in Appendix B (L859-L905) under Neural SDF evolution (see Figure 9). The results show that AIRe can be also applied in this PDE-driven INR setting.
>
> **Training time and GPU consumption.**
>
> We report both the training time and GPU utilization during the image-fitting task (see Table _Training Time vs. GPU Usage (DIV2K)_ below). The observed increase in training time arises from an implementation detail: pruning is done via masking weights rather than outright removing them. Since PyTorch does not support removing individual weights from the computational graph using `torch.no_grad`, masked neurons continue to participate in both the forward and backward passes. Consequently, pruning introduces additional computational overhead during training. (But do note that this is mainly due to PyTorch limitations, and not intrinsic to our framework.)
>
> Importantly, this overhead disappears after training. Once pruning is complete, the unused neurons are completely removed from the model, resulting in significantly faster inference and lower GPU consumption (see Table 9). This improvement is particularly valuable in practice, as it allows larger batch sizes and more efficient deployment in GPU-intensive applications.
>
>
> _Table 9: Inference performance for the image task._
> |  | **CPU time (s)** | **GPU avg usage (%)** | **GPU inference time (×10⁻⁴ s)** |
> |---|---|---|---|
> | Large | 0.74 | 44.17 | 4.324 |
> | AIRe | 0.61 | 31.53 | 3.193 |
> | Improvement | 17.16% faster | 28.62% lower | 26.17% faster |
>
>
> _Training Time vs. GPU Usage (DIV2K)_
> | **Model** | **Metric** | **Large** | **AIRe** |
> |-------|--------|----------|-------|
> | SIREN | Training Time (s) | 693.27 | 924.38 |
> | SIREN | GPU Usage (%) | 94.11 | 93.60 |
> | FINER | Training Time (s) | 823.00 | 1081.00 |
> | FINER | GPU Usage (%) | 94.05 | 94.85 |
>
> **Comparison with other baselines**
>
> Thank you for pointing out the relevance of LosslessINR [2] as an additional baseline. We agree that methods targeting compact or equivalent implicit representations provide valuable context for evaluating pruning techniques. While we did not include LosslessINR specifically -- primarily because its formulation targets lossless structural transformations rather than neuron-level pruning -- we did incorporate a closely related and widely adopted compression baseline, SPDER [3]. The results in the DIV2K dataset are presented in Table 8.
>
> _Table 8: AIRe applied to SPDER architecture._
> | **Model (Images)** | **Variant** | **PSNR** | **Size reduct.** |
> |---|---|---|--|
> | SPDER | Large | 30.77 | – |
> | SPDER | Ours | 34.61 | 34.89% |
>
> Our training achieves better quality (~4dB) with a reduction of 34.89% of parameters, showing the ability of AIRe to adapt different architectures to the given signal.
>
> [1] Novello et al., *Neural Implicit Surface Evolution*, ICCV 2023.
>
> [2] Han, Woo Kyoung, et al. *Towards Lossless Implicit Neural Representation via Bit Plane Decomposition.* Proceedings of the Computer Vision and Pattern Recognition Conference. 2025.
>
> [3] Shah & Sitawarin, *SPDER: Semiperiodic Damping-Enabled Object Representation*, ICLR 2024.

---

### Official Review · Reviewer_Gj4a · 2025-11-01

**Soundness:** 3
**Presentation:** 3
**Contribution:** 2
**Rating:** 4
**Confidence:** 3

**Summary:**

The paper addresses the challenges of parameter redundancy and limited representational capacity in implicit neural representations (INRs). It proposes AIRe (Adaptive Implicit neural Representation), an adaptive training framework that alternates between neuron pruning to remove redundant units and frequency densification to introduce additional input frequencies where the signal underfits. This process yields compact networks that better balance efficiency and reconstruction quality.

The method is validated on several signal representation tasks, including image reconstruction, 3D surface fitting, and neural rendering (NeRF). AIRe consistently matches or surpasses larger overparameterised baselines in reconstruction accuracy while using substantially fewer parameters, and outperforms general-purpose adaptive and pruning methods  (RigL, DepGraph). However, the gains are less pronounced on complex, practically relevant tasks such as NeRF-based view synthesis, where improvements over smaller fixed models remain marginal and size reduction benefits limited. Future revisions could strengthen the work by discussing the intended applicability of the approach beyond signal fitting, analysing inference-time efficiency, and broadening or clarifying comparisons with existing INR-specific pruning and sparsity methods.

**Strengths:**

**S1. Exploration of INR-specific network pruning.** The paper tackles a timely and important problem—adapting pruning strategies to the unique characteristics of implicit neural representations (INRs). This direction is both compelling and relevant, as multilayer perceptrons (MLPs) remain a major computational bottleneck in tasks such as neural rendering. The proposed approach demonstrates clear benefits over general-purpose pruning methods (Table 2), showing consistent and INR-aware improvements in efficiency–accuracy trade-offs.

**Weaknesses:**

**W1. Limited performance gains on relevant or real-world tasks.** The most extensive experiments (Table 8, supplementary) show only marginal improvements on the NeRF reconstruction task. The proposed method achieves roughly 20 % model-size reduction but delivers only minor accuracy gains over the same-size model trained from scratch. This raises questions about its effectiveness for complex, practically relevant scenarios such as neural rendering.

**W2. Missing analysis of inference efficiency.** While the paper discusses reductions in model size and training time (Table 7, supplementary), it does not analyse the effect of pruning on inference time, a major bottleneck for INRs in applications like real-time rendering and novel-view synthesis. Understanding how architectural adaptation impacts forward-pass latency is essential for assessing practical utility.

**W3. Incomplete comparison with prior INR pruning work.** The omission of Zell et al. (2022) from quantitative comparison is insufficiently justified. Although that method reduces model size only through input-layer pruning, it remains—by the authors’ own admission—“the only prior work exploring the pruning (or adaptation) of INRs” (l. 118–120) and should be included for completeness. Moreover, other relevant studies addressing sparsity or compression in INRs [1–2] are not discussed; establishing their relation to the proposed approach would clarify the work’s novelty and scope.


References

[1] Lee, J., Tack, J., Lee, N., & Shin, J. Meta-learning sparse implicit neural representations. NeurIPS 2021.

[2] Jayasundara, D., Rajagopalan, S., Ranasinghe, Y., Tran, T. D., & Patel, V. M. (2025). SINR: Sparsity-Driven Compressed Implicit Neural Representations. CVPR 2025.

**Questions:**

**Q1. Applicability to real-world INR tasks.** If the improvements on neural rendering (e.g., NeRF) remain minor, where do the authors envision this approach having the greatest practical impact? In which INR domains or signal types does adaptive pruning most clearly translate into meaningful efficiency or accuracy gains?

**Q2. Inference efficiency.** What is the impact of pruning and architectural adaptation on inference time—particularly for forward-pass latency in real-time or high-resolution INR settings—beyond the reported reductions in model size and training time?

**Q3. Comparison with prior INR pruning methods.** The paper would benefit from a deeper discussion of related INR pruning and sparsity works (e.g., Zell et al., 2022; Lee et al., 2021; Jayasundara et al., 2025). If feasible, these could be included as additional baselines; otherwise, it would be helpful to clarify why such comparisons were not performed and how the proposed approach conceptually differs from them.

---

> ### Author Response · Authors · 2025-11-26
>
> We appreciate the reviewer’s valuable feedback. It is encouraging that they find our problem important and timely, and consider our work direction as compelling and relevant.
>
> **Applicability to real-world INR tasks.**
>
> The main NeRF evaluation is presented in Table 1 (left), where AIRe achieves substantial model-size reduction while preserving or even improving reconstruction quality (PSNR). This shows that AIRe can adapt the network architecture to the input data in the NeRF setting.
> The additional experiment in Table 11, in contrast, fixes the final architecture and compares AIRe-trained models with the same final architecture trained from scratch. Even in this homogeneous setup, the adaptive training dynamics of AIRe still deliver consistent improvements, showing that the gains are not solely due to architectural resizing.
> Finally, we emphasize that surface reconstruction via neural SDFs is one of the most important applications for INRs, and it has been integrated with modern 3DGS pipelines (e.g., GS-Pull, SuGaR).
>
> **Inference efficiency.**
>
> We agree that pruning should not only reduce model size but also yield measurable improvements in inference speed. To address this, we evaluated the forward-pass latency and computational cost (GFLOPs) for the large SIREN model and our pruned architecture (AIRe) on the trained NeRF models in Table 1. The results are summarized in Table 10:
>
> _Table 10: Inference efficiency comparison_
> | **Variant** | **Gflops**  |
> |--|--|
> | Large | 7.49 |
> | AIRe | 5.17 |
> | Improvement | 30.96% |
>
> These results show that computational cost is reduced by ~31%, consistent with the structural sparsity introduced by our pruning mechanism. This indicates that removing neurons and reducing FLOPs directly accelerates the forward pass on GPU. Overall, these measurements confirm that our architectural adaptation yields meaningful improvements in GPU inference latency, which is a critical metric for real-time or high-resolution INR workloads.
>
>
> **Comparison with prior INR pruning work.**
>
> Thank you for your comment; we have implemented the method of Zell et al. [3] and added a quantitative comparison with AIRe using the DIV2K dataset (Table 2). The results are as follows:
>
> _Table 2 (partial): Comparison of AIRe with Zell et al. [1]_
> | Method      | PSNR             | SSIM          |
> |-------------|------------------|----------------|
> | Zell et al. | 18.96 ± 1.92     | 0.39 ± 0.08    |
> | AIRe        | **37.53 ± 3.75** | **0.96 ± 0.02** |
>
> Details can be found in the "Comparison against existing pruning baselines" paragraph in the main text (L290-L307). Overall, these results indicate that AIRe yields substantially higher reconstruction quality while maintaining stable and efficient training.
>
> As for SINR [2] and Meta-SparseINR [3], we found their inclusion to be unfortunately infeasible; SINR [2] has no official implementation available, and Meta-SparseINR was too costly to run due to its requirement of pre-training on the entire dataset (which takes several hours) before fitting an INR to a single signal (our method does not require anything close to such a pretraining phase).
>
> [1] Benbarka, Nuri, Timon Höfer, and Andreas Zell. *Seeing implicit neural representations as fourier series.* Proceedings of the IEEE/CVF Winter Conference on Applications of Computer Vision. 2022.
>
> [2] Jayasundara, D., Rajagopalan, S., Ranasinghe, Y., Tran, T. D., & Patel, V. M. (2025). *SINR: Sparsity-Driven Compressed Implicit Neural Representations.* CVPR 2025.
>
> [3] Lee, J., Tack, J., Lee, N., & Shin, J. *Meta-learning sparse implicit neural representations.* NeurIPS 2021.

---

### Author Response · Authors · 2025-12-03
**Final remarks**

We thank the Reviewers and the ACs for their thoughtful and constructive evaluations. We have addressed all questions and concerns raised during the review process and updated the manuscript accordingly. We are encouraged by the reviewers’ remarks indicating that the revisions resolved their earlier concerns. Below, we summarize the key improvements.

We clarify that AIRe targets practical INR settings such as image fitting, SDF reconstruction, and NeRF, where our automatic approach for defining the network architecture enables a strong reduction in model size while preserving or even improving quality (Gj4a).

We evaluate the efficiency of inference with models trained by AIRe (in terms of GFLOPs and latency) and CPU/GPU timing for image fitting (Gj4a, aRcV). AIRe’s pruned networks reduce FLOPs by 31% and yield 17–26% faster GPU inference with 29% lower GPU memory usage. We also explain that the training-time overhead (Bn28) is due to PyTorch’s masked-pruning mechanism; inference, in contrast, strictly benefits from the compact architecture.

We compared against Zell et al., a key pruning method targeted to INRs, and observed large PSNR gains (Gj4a, aRcV). We also integrated AIRe with SPDER (aRcV, Bn28), improving PSNR by 4 dB while reducing parameters by 35%. Additionally, we clarify the conceptual relation between pruning/densification in INRs and recent Gaussian-based representations (aRcV).

To address comparisons with other tasks, such as PDEs, we added a mean-curvature–flow experiment showing that AIRe adapts a 4D SDF without degrading accuracy (Bn28). We also improved the explanation of densification (why $2\cdot\omega_j$?), showing that after densification most of the novel neurons are highly contributive to the signal, clarified that in some experiments we prune both input and hidden neurons for all tasks, justified the fixed threshold ($\epsilon = 0.01$) via Thrm 2, and added in Fig. 10 spectral visualizations and convergence curves (HQzk). Several figures were additionally revised for clarity (HQzk).

---

### Meta-Review · Area_Chair_Xd29 · 2025-12-29

**Summary:**

The paper makes a solid contribution to INR methodology, but the limited impact on NeRF and restriction to low-dimensional signals cap the significance. The core idea of combining pruning (with a targeted weight decay mechanism) and frequency densification to adaptively adjust INR architectures is well-motivated and technically sound. The targeted weight decay mechanism is the most interesting contribution—it allows information to be transferred from low-contributing neurons before they are removed, which the ablations show is critical for maintaining reconstruction quality.

The experimental evaluation is reasonably thorough, covering images, SDFs, and NeRFs. The results on image fitting and SDF reconstruction are convincing, with AIRe achieving comparable or better quality to large networks while using significantly fewer parameters (up to 84% reduction for SDFs). The added comparison with Zell et al. establishes clear superiority over the only prior INR-specific pruning method. However, the NeRF results remain modest—roughly 20% parameter reduction with marginal quality improvements—which limits the practical significance for neural rendering applications where such methods would have the most impact.

**Reviewer Concerns:**

The inference efficiency concern raised by Gj4a was fully resolved in the rebuttal. The authors demonstrated 31% FLOP reduction and 17-26% faster GPU inference times, confirming that the pruned architectures yield practical speedups at deployment time.
The missing comparison with Zell et al. was addressed with new experiments showing AIRe achieves 37.53 PSNR compared to 18.96 for Zell et al. on DIV2K, establishing clear advantage.

Reviewer HQzk's questions about the 2ωj frequency choice were clarified through reference to Theorem 1 and Bessel function properties. The fixed pruning threshold ε=0.01 is justified via Theorem 2's stability bound. The requested spectral visualizations were added as Figure 10.

The SPDER comparison requested by aRcV was provided, showing AIRe improves PSNR by 4dB while reducing parameters by 35%, demonstrating the framework's applicability to other INR architectures.

The concern about marginal NeRF performance was partially addressed by noting that SDF reconstruction is itself a significant INR application. However, the limited gains on NeRF remain a weakness—the method appears most beneficial for simpler signal fitting tasks.

The training overhead acknowledged by the authors is a design limitation, though the inference benefits are clear and likely outweigh this for deployment scenarios.

**Reviewer Scores:**

Reviewer Gj4a: 4 → 5. Both major concerns (inference efficiency, Zell et al. comparison) were comprehensively addressed with new experiments.

Reviewer Bn28: 6 → 6. Already positive; concerns about PDEs and training time were addressed satisfactorily.

Reviewer HQzk: 4 → 5. Explicitly raised rating after rebuttal, stating concerns were resolved.

Reviewer aRcV: 4 → 5. Explicitly raised rating, stating all concerns were adequately addressed.

---

### Decision · Program_Chairs · 2026-01-26

Reject